# Fabrication and Applications of Ceramic-Based Nanofiber Materials Service in High-Temperature Harsh Conditions—A Review

**DOI:** 10.3390/gels9030208

**Published:** 2023-03-09

**Authors:** Jing Zhang, Xi Zhang, Lifeng Wang, Junxiong Zhang, Rong Liu, Qilong Sun, Xinli Ye, Xiaomin Ma

**Affiliations:** 1School of Textile and Clothing, Nantong University, Nantong 226019, China; 2Nantong Sanzer Precision Ceramics Co., Ltd., Nantong 226001, China; 3School of Civil Aviation, Northwestern Polytechnical University, Xi’an 710072, China; 4National Equipment New Materials and Technology (Jiangsu) Co., Ltd., Suzhou 215101, China

**Keywords:** high-temperature, ceramic, nanomaterials, applications

## Abstract

Ceramic-based nanofiber materials have attracted attention due to their high-temperature resistance, oxidation resistance, chemical stability, and excellent mechanical performance, such as flexibility, tensile, and compression, which endow them with promising application prospects for filtration, water treatment, sound insulation, thermal insulation, etc. According to the above advantages, we, therefore, reviewed the ceramic-based nanofiber materials from the perspectives of components, microstructure, and applications to provide a systematical introduction to ceramic-based nanofiber materials as so-called blankets or aerogels, as well as their applications for thermal insulation, catalysis, and water treatment. We hope that this review will provide some necessary suggestions for further research on ceramic-based nanomaterials.

## 1. Introduction

In recent years, ceramic-based nanofiber materials have received more and more attention, and the publication of articles on ceramic-based nanofiber materials is also a growing trend. Regarding traditional ceramic materials, their brittleness greatly limits the development of their application areas, and flexible ceramic-based nanofibers are an efficient approach to improving this shortcoming. With the research and the derivation of many types of ceramic materials, these ceramic materials, because of their low density, low thermal conductivity, good chemical stability, and many other superior properties, are widely used in the field of high-temperature insulation, acoustics, catalysis, and other fields.

In this paper, we introduce ceramic-based nanofiber materials from two perspectives, the preparation of ceramic-based nanofiber materials and their application fields (as shown in Figure 1), in which their preparation is divided according to conventional ceramic fiber materials and ceramic aerogels. From these classifications, we describe the properties, chemical compositions, advantages, and disadvantages of the prepared materials in detail. In terms of applications, we highlight applications in high-temperature thermal insulation, air filtration, water treatment, sound absorption, electromagnetic wave insulation, battery separators, catalytic applications, etc. Finally, we offer an outlook on ceramic-based nanofiber materials.

## 2. Ceramic Nanofibers

Conventional ceramic materials are usually sintered from natural raw materials such as clay and quartz, typical silicate materials. Their main elements are silicon, aluminum, and oxygen. These common ceramics are abundant sources and of low cost, and the mature process makes it easier to obtain the material. The composition can be divided into aluminosilicate, alumina, zirconia, quartz, glass fiber, etc.

### 2.1. Oxide Fibers

#### 2.1.1. Al_2_O_3_ Fiber

Aluminosilicate, alumina, and mullite mainly consist of aluminum–silicon refractories, but the content of Al_2_O_3_ and the content of the main solid-phase composition are different.

Aluminosilicate belongs to a form of silicate whose composition is Al_2_O_3_ and SiO_2_, and the higher the Al_2_O_3_ content, the better the heat resistance, as can also be seen in Table 1 [1]. These materials are excellent in terms of oxidation resistance, creep resistance, and high-temperature resistance, as well as thermal stability [2], and the high-temperature resistance reaches up to 1000–1500 °C for different chemical compositions; these mineral compositions can be divided into several categories of aluminosilicate fibers, as seen in Table 1 [1]. The first four categories, when placed at 1000 °C for a long time, will precipitate mullite, square quartz, and other crystals, but the corresponding fiber will become brittle, affecting the service life. The last two categories are crystalline fibers themselves, which are used at higher temperatures.

Aluminum silicate can be used as a heat insulation material for engine exhaust pipes and automobile silencers, the electric insulation of electric furnaces, metal melting, and high-temperature Karma high-pressure gas-filtering materials, etc. [3,4]. Aluminum silicate powder is often used as a fireproof material; for example, in the glass industry, it is used to build glass kilns, which indicates that its development prospects are very broad. In addition, it has been shown that the excellent electrochemical properties of this material can be used to prepare composite materials in the future, to further exploit its application value [5].

Alumina is another of the most important functional materials among inorganic materials, and it is one of the most researched materials nowadays. The main components of alumina fibers are Al_2_O_3_ and a small amount of SiO_2_, which can stabilize the crystal and inhibit its growth. Alumina fibers are ultra-lightweight high-temperature insulation materials that have received much attention at home and abroad. Jia [6] summarized that there are many materials that can be processed into Al_2_O_3_ fibers. Alumina fibers have low thermal conductivity, low heating shrinkage, good chemical stability, good high-temperature resistance, and can be used at temperatures up to 1450–1600 °C, which is higher than ordinary aluminosilicate fibers, used as thermal insulation materials for nuclear reactors and space shuttles. In addition, they can also be used as catalyst carriers for the chemical industry. Their chemical resistance is also excellent for use in the fields of environmental protection and recycling technology.

The company 3M has developed many Nextel series alumina fibers, the earliest of which was the Nextel-312 fiber [7]. The B_2_O_3_ component in this fiber was initially intended to increase the nucleation density of mullite and reduce grain growth [8], but it was found that B_2_O_3_ volatilizes easily at high temperatures, resulting in insufficient thermal stability in Nextel-312. The Nextel-440 fiber, which has better high-temperature resistance, was prepared for this B_2_O_3_ reduction component. Later, the Nextel-550 fiber completely removed B_2_O_3_, providing it with good high-temperature resistance and creep resistance compared to the previous two fibers. The development of the Nextel-610 and Nextel-720 fibers has proven that not all oxides have low tensile strength and resistance to high-temperature creep [9,10,11]. The Nextel-650 [12,13] fiber content includes ZrO_2_, a small amount of Y_2_O_3_, and a very small amount of Fe_2_O_3_, in addition to Al_2_O_3_. The Nextel-650 fiber has a monofilament tensile strength of 2.5–2.7 GPa; it hardly creeps at 1100 °C and exhibits good creep resistance [14]. In addition, 3M has found that the characteristics of fiber continuity can be used to prepare filter carriers with a porous network structure [15]. Jia [16] prepared flexible α-Al_2_O_3_ nanostructured fibers using alumina and isopropanol as raw materials, glacial acetic acid and hydrochloric acid as catalysts, and polyvinyl pyrrolidone (PVP) as a spinning auxiliary. The characterization results showed that the fiber surface was dense, without holes and cracks. Its mechanical properties also showed good data. It was also found that the phase structure and morphology of alumina nanostructure fibers doped with SiO_2_ changed significantly. The γ-Al_2_O_3_ fiber has a minimum thermal conductivity of 0.0476 W·m^−1^·K^−1^ and is converted to α-Al_2_O_3_ after calcination, and its thermal conductivity increases to 0.0773 W·m^−1^·K^−1^. Zhang [17] also used sol–gel combined with electrospinning to prepare flexible alumina nanostructured fibers. It was found that the addition of MgSO_4_ could initially delay the phase transition of fibers from γ-Al_2_O_3_ to α-Al_2_O_3_, and then effectively promote sintering. When subjected to heat and stress–strain, the fibers exhibited good thermal stability and flexibility. Doped CaO and SiO_2_ two-phase additives will cause the fiber to possess lower thermal conductivity, which has better applications in the field of thermal insulation.

Li [18] used aluminum powder as an aluminum source, tetraethyl orthosilicate as a silicon source, and polyethylene oxide as a presumptuous auxiliary, and prepared alumina fibers by sol–gel combined with electrospinning technology. Compared with pure alumina fiber, this flexible alumina nanostructured fiber has a dense surface without holes and cracks, showing good flexibility and impressive mechanical properties. Its tensile strength is increased by 78%. In addition, its filtration efficiency is as high as 99.88%, indicating its good application prospects in high-temperature filtration and other fields. Wei [19] studied the heat treatment process of sol–gel to prepare alumina fibers. It was found that the heat treatment rate had a great influence on the performance of the fiber, and it is not suitable to use a high heating rate. In the heat treatment at 550 °C, the volume and mass shrinkage of alumina fibers accounted for a high proportion (up to 85%). In heat treatment with a temperature above 800 °C, the crystal phase will change from amorphous to γ-Al_2_O_3_ to mullite.

On the basis of alumina fibers, ultra-fine alumina fibers can be obtained by reducing the diameter of alumina fibers. Milanović [20] used electrospinning to prepare alumina fibers, adding magnesium oxide to the alumina structure for sintering. The sintered fiber is finer in diameter and smaller in volume, so as to show better flexibility and specific surface area. In addition, the design of a certain unique structure can also be used to significantly improve the performance of ultra-fine alumina fibers [21].

Almeida [22] studied a CeraFib75 alumina continuous fiber containing 175 μm mullite particles and trace γ-alumina. Compared to the previously developed Nextel-720 fiber (Nextel-720 fiber was once considered the commercial Al_2_O_3_ fiber with the best resistance to high-temperature creep), it can exhibit higher strength retention at high temperatures. Song [23] used a sol–gel method to synthesize alumina fibers with two systems: AlCl_3_-Al powder–H_2_O and Al(NO_3_)_3_-Al powder–H_2_O. In the aluminum chloride system, an Al/AlCl_3_ molar ratio between 2 and 5 can lead to a better sol, while, for the aluminum nitrate system, the spinning range is narrow. It was found that Cl^-^ is more difficult to decompose than NO_3_^-^, but the phase transition temperature from γ-Al_2_O_3_ to α-Al_2_O_3_ in the AlCl_3_-Al powder–water system is lower. When sintered above 800 °C, the fibers of the aluminum nitrate system are less broken and have fewer cracks than the fibers of the aluminum chloride system. Sun [24] synthesized an Al_2_O_3_-SiO_2_ precursor sol with a certain viscosity with aluminum nitrate, isopropanol, and ethyl orthosilicate. After the hydrolysis polycondensation reaction, 10–20 nm sol particles would be generated in the sol system. In acidic conditions, Si-O-Si, Si-O-Al, Al-O-Al, and other structural units existed in the sol extraction product. Under aging, the system changes from sol to gelatinous, which is reversible and stable.

Chen [25] mixed alumina sol and silica sol, added spinning additives, and then obtained continuous alumina-based ceramic fibers by dry spinning and pyrolysis, and sintering. It was found that the ratio of aluminum powder to aluminum chloride, aluminum salt concentration, reaction time, oxygen content, and other factors had an effect on the sol. The pH value of the solution is a key factor in determining the performance of the aerosol. Tan [26] mixed acetic acid, ethyl orthosilicate, a sol stabilizer, an inorganic salt of aluminum, and aluminum powder, and reacted them at a certain temperature, with continuous stirring and condensation reflux, to obtain a precursor silicon–aluminum sol with good stability. Spinning additives were added to obtain a spinnable precursor sol. Alumina-based fibers that could be used in aerospace, automobiles, and composite reinforcements can be obtained through dry/wet spinning, drying, and sintering. At the same time, Tan also mixed aluminum nitrate and PVP with three different organic acids—lactic acid, malic acid, and tartaric acid, respectively—to prepare Al_2_O_3_-based ceramic continuous fibers. Experiments showed that the solution spinning performance was the best when the mass ratio of aluminum nitrate, organic acid, and PVP was 10:3:1.5, without a dependency on the type of organic acid.

Mahapatra [27] prepared α-Al_2_O_3_ microfibers with a diameter between 100 and 500 nm using aluminum acetate sol and PVP polymer solution as raw materials. The surface morphology and fiber diameter of the alumina fibers were measured. Characterized by crystal equality, it had a surface area of 40 m^2^/g and was highly crystalline in nature. Wang [28] successfully prepared a new independent γ-alumina fiber film with good flexibility, randomly arranged by nanofibers by electrospinning. This alumina fiber film possessed great thermal stability (up to 900 °C) and high tensile strength. In addition, its filtration efficiency could be as high as 99.97%, indicating its potential in the field of high-temperature filtration.

By studying the preparation method of alumina, Li [29] found that by adjusting some process parameters, they could successfully design alumina fibers with different diameters in the sol–gel process, with different components and different structures, as shown in Figure 2. In addition, inorganic aluminum sources with a controlled number of organic components, an optimized calcination temperature and speed, etc., could be used to save costs and improve the overall performance of fibers.

Wang [30] successfully synthesized mesoporous alumina fibers with a surface area of 264.1 m^2^·g^−1^ using electrospinning and sol–gel methods. Calcined fibers exhibited an adsorption capacity of up to 781.25 mg·g^−1^. Li [31] combined the sol–gel method and heat treatment process to obtain an alumina fiber mat with good flexibility that can be applied to high-temperature processes, filtration, catalysis, and other fields. By adjusting the parameters, the diameter of the alumina fiber can be controlled to 1~3 μm. Sedaghat [32] prepared an alumina mat by sol–gel centrifugal spinning using AlCl_3_·6H_2_O, Al powder, and SiO_2_ as raw materials. The phase change temperature of this mat is 600 °C, and there is no amorphous phase at 800 °C, while *θ*-alumina is the most notable component. The transition of α-Al_2_O_3_ is completed at 1200 °C, and the presence of SiO_2_ inhibits the transition of α-Al_2_O_3_. The optimal proportion of Si in the alumina mat is 4%, and, if it exceeds this, mullite will form in the microstructure, as displayed in Figure 3. Akia [33] optimized the preparation of aluminum isopropyl and poly(vinyl alcohol) fine fibers, and calcined the fiber samples at different temperatures to obtain γ and α phases. The analysis results showed that the surface area of the mesoporous γ-Al_2_O_3_ structure was 261 m^2^·g^−1^, and the average diameter of the crystalline alumina fine fibers was 272 nm.

The amount of alumina in mullite is between that of Al_2_O_3_ fibers and aluminum silicate fibers, fluctuating between 72 and 78%. Mullite is a high-quality refractory material, mainly including high-purity electrofused mullite, ordinary electrofused mullite, all-natural bauxite concentrates, sintered mullite, and lightly burned mullite. Mullite is formed when aluminosilicates are artificially heated.

Chen [34] prepared mullite ceramic fibers with aqueous solutions of aluminum nitrate, isopropyl alumina, and tetraethylchlorosilicate. Spinnable sols exhibited high-viscosity shear thinning flow behavior during the gelation time. By temperature regulation, the degree of gelatinization could be stabilized at a certain value. Crack-free ceramic fibers with a tensile strength of approximately 900 Mpa and a diameter between 45 and 50 μm would be formed in the sintering conditions under 1100 °C. Zadeh [35] synthesized continuous mullite nanofibers using sol–gel and electrospinning techniques. Adjusting the content of polyvinyl butyral (PVB) can obtain materials of different qualities. When the PVB content is between 4 and 6%, after calcination at 1200 °C, very pure, smooth, uniform mullite nanofibers can be obtained. Moreover, the mullite nanofibers have a small diameter (85~130 nm). Song [36] prepared mullite using the same method. Due to densification, the average diameter of the fibers decreases from 318 to 261 nm when the temperature rises from 800 to 1200 °C. The surface of the mullite fibers becomes relatively smooth at 1000 °C due to the viscous flow of amorphous silica. At the same time, the fiber exhibits uniform characteristics at an Al/Si molar ratio of 2.98. The low modulus of elasticity and good flexural properties around 25.18 GPa reveal excellent flexibility. da Costa Farias [37] produced sub-micro-mullite fibers for the first time using solution blow spinning technology. This technology is not only effective but also has high yields. This new method produces mullite without cracks, with an average diameter of 800 nm and a high surface area.

#### 2.1.2. ZrO_2_ Fiber

ZrO_2_ ceramics are also a potential material for high-temperature applications due to possessing excellent properties such as high melting and boiling points, high hardness, low thermal conductivity, high toughness, wear resistance, etc. Compared with other fibers, such as mullite, alumina, and alumina-silicate, ZrO_2_ fibers could still maintain their integrity at a higher temperature of up to 2200 °C. At the same time, they have the advantages of corrosion resistance, oxidation, non-pollution, etc., and they are currently the most popular international refractory material.

There are three crystalline forms of pure ZrO_2_ at atmospheric pressure: monoclinic (m-ZrO_2_), tetragonal (t-ZrO_2_), and cubic zirconia (c-ZrO_2_) [38]. These three crystallines exist in different temperature ranges and can be transformed into each other. When the ZrO_2_ matrix is lowered from a high temperature to 900 °C, the tetragonal form will transform into the monoclinic form with volume expansion, cracks, and residual stresses, which appear inside the matrix and are prone to brittle fracture [39]. Rare earth oxides such as Y_2_O_3_ and CeO_2_ are therefore usually added to inhibit the fracture caused by the reverse transformation.

Han [40] examined thermal insulation tiles composed of short-cut zirconia fibers in an arc wind tunnel, which showed that zirconia can withstand very high temperatures. Similarly, zirconia fibers have some defects in their mechanical properties, and usually researchers add alumina to improve their properties. Li [41], using slurry impregnation and discharge plasma sintering techniques, prepared alumina-reinforced zirconia composites, and the toughening effect was nearly doubled, as also confirmed by Wang [42]. Lang [43] also prepared YSL composites with excellent properties based on this, which greatly enhanced the prospects of zirconia.

Wang [44] synthesized a variety of three-dimensional sponges based on oxide ceramics using an economic and effective blowing and spinning technology. Among them, the ZO_2_ nanofiber sponge not only showed high-temperature resistance, but also showed excellent high-energy absorption and good compressive strain and recovery, and it can be used in high-temperature applications, catalysis, electricity, and other fields. Sun [45] prepared and characterized zirconium polyacetate (PZA) and found that the obtained cubic-phase ZrO_2_ fibers, after adding 6 mol% Y_2_O_3_ and calcination at 1200 °C, possessed a grain production activation energy of approximately 24.18 kJ·mol^−1^. Moreover, the quality of the fiber can be further improved by spinning and heat treatment of the sol–gel of the aforementioned PZA. By using ZrOCl_2_·8H_2_O (ZOC) and H_2_O_2_ as raw materials, Liu [46] proposed a novel inorganic sol–gel method to prepare high-quality ZrO_2_. It was found that the inorganic polyzirconium molecules in the spinning solution had a double-stranded structure, which resulted in the excellent strength and flexibility of the prepared ZrO_2_ fibers.

Wang [47] reported a nanofiber sponge based on yttrium-stabilized ZrO_2_ (YSZ), exhibited in Figure 4. Porous three-dimensional sponges composed of YSZ nanofibers are not only lightweight but also elastic, both at room temperature and high temperatures. This sponge has great application potential in the field of filtration, with filtration efficiency of up to 99.4%.

#### 2.1.3. Glass Fiber

Glass fiber is an inorganic non-metallic material with excellent performance. It has the advantages of thermal insulation, heat resistance, and great mechanical properties, as well as some disadvantages, including brittleness, poor wear resistance, and a monofilament diameter of a few microns to twenty microns. Therefore, glass fiber is also often used as a reinforcing material for composite materials, electrical insulation materials, and thermal insulation materials.

Ma [48] showed, in their research experiments on carbon and glass fiber prepregs, that glass fibers do not decompose at 850 °C compared to carbon fibers. Both of them have an inhibitory effect on the pyrolysis of the composites, which is more pronounced for glass fibers. Sujatha [49] examined two different fiber glasses. One was an alkali-resistant glass fiber, and the other was an electronic-grade glass fiber. Through experiments, it was found that the random addition of these two fibers can improve the unlimited compressive strength and energy absorption capacity of reinforced soil.

Wang [50] prepared glass fiber by using a high-temperature furnace and auxiliary raw materials. Through the test analysis, they found that the diameter of the glass fiber obtained by this method was 11.28 μm. The average linear density value reached 4.42 g/km, and the average tensile strength of the glass fiber was 0.57 N. In addition, glass fiber also has good light absorption performance, and the acid and alkali resistance are significantly improved compared to those of unmodified glass fiber. Ma [51] found that glass fiber can improve the shortcomings of recycled concrete with many internal holes, poor compressive resistance, and poor acid and alkali resistance.

Quartz, mainly consisting of SiO_2_, is one of the most widely distributed minerals on the Earth’s surface. It was used in early times to produce stone axes and other tools for survival and was later fused to create glass, which is also used in various areas, including metallurgy, construction, and the chemical industry. Quartz fiber is an inorganic fiber composed of high-purity quartz or natural crystal, generally a few microns to tens of microns in diameter. Quartz fiber is an excellent material for high-temperature resistance and is often used as a reinforcing phase for some composite materials. The SiO_2_ content in quartz fiber is almost 100%. Its high-temperature resistance is higher than that of silica oxygen. In addition, quartz has a long-term usage temperature of up to 1200 °C. It also has superior electrical insulation, ablation resistance, and chemical stability, which is one of the reasons that quartz fiber is used in national defense, the military, aerospace, and other fields.

As mentioned above, oxide fibers have shown their potential for applications in high-temperature and harsh oxidative conditions, because of their excellent oxidation resistance and chemical stability. However, there exist shortcomings that have prevented their utilization in the grain-coarsening process over 1300 °C. Therefore, a new goal is to search for novel materials for higher-temperature utilization, such as carbide or nitride.

### 2.2. Nitride Fibers

#### Si_3_N_4_ Fiber

The most widely used among nitride ceramic fibers is silicon nitride (Si_3_N_4_). Si_3_N_4_ fiber is a type of ceramic fiber with high-temperature resistance and high strength. The high-temperature resistance of Si_3_N_4_ fibers is particularly outstanding. The maximum use temperature can reach 1300 °C in oxidizing gases and up to 1800 °C in non-oxidizing gases. In addition, its mechanical and physical properties are also excellent. Based on the many excellent properties of Si_3_N_4_ fiber, material scientists in various countries are very interested in it. Si_3_N_4_ ceramic fibers have also undergone a very extensive research process. Thus far, the research on Si_3_N_4_ ceramic fiber has achieved great results.

Zhang [52] synthesized a novel silicon nitride nanowire sponge. They used carbon-doped silica sol foam as the skeleton structure and prepared it via a simple and effective carbon thermal reduction reaction. The silicon carbide nanowire microspheres prepared by this method have a uniformly curved morphology and high specific surface area due to the stable growth environment. Based on the various characterizations of silicon carbide nanowire sponges, it can be found that it provides new prospects for filtration, heat preservation, catalysts, and other fields. A schematic diagram of its microsphere formation is shown in Figure 5. Huo [53] sintered silicon nitride nanofiber knitted ceramic foam by an in situ reaction on the basis of a stable silicon foam. This new silicon carbide ceramic foam consists of a three-dimensional nanofiber assembly network with a diameter of 15–100 nm. The preparation method is simple and low-cost, which opens up a new opportunity for catalytic adsorption and high-temperature heat insulation and other fields.

Compared to oxide fibers, nitride fibers displayed higher utilization potential in high-temperature conditions up to 1800 °C. However, due to their worse oxidation resistance, they are not suitable for utilization in oxidation conditions over 1800 °C for long periods. Therefore, it is necessary to choose a non-oxidized fiber covered with oxide to improve its high-temperature utilization, among which carbide is an excellent choice.

### 2.3. Carbide Fibers

#### SiC Fiber

Silicon carbide ceramic fiber is a typical example of a carbide ceramic fiber. In addition to good mechanical properties, silicon carbide ceramics also have high bending strength, excellent oxidation resistance, good corrosion resistance, high wear resistance, and a low friction coefficient. In addition, the high-temperature mechanical properties of silicon carbide fibers are the best among known ceramic fibers. Silicon carbide fiber, which has many excellent properties, has become an important material in the field of high-end equipment and technology.

In the course of the research, it has been found that it is very challenging to produce high-performance materials that can be put to use. Most are hindered by chemical corrosion, low thermal stability, and complex preparation processes. Liu [54] reported a novel free-standing hollow SiC fiber pad, which is flexible and acid- and alkali-resistant. The preparation process is divided into three steps: preparation of a shell fiber, curing, and pyrolysis. The multilayer scattering mechanism of n-doped hollow silicon carbide fibers is shown in Figure 6. The obtained hollow structure has a cavity wall thickness of 1.5 μm. After analyzing the morphology, composition, etc., of this hollow SiC fiber, it was found that it has the advantages of good flexibility, high thermal stability, corrosion resistance, a light weight, and low thermal conductivity. These excellent characteristics mean that the SiC fiber pad has very important applications in high-temperature thermal insulation, high-temperature catalysis, and other fields.

Li [55] successfully prepared microscale silicon carbide fiber pads via the blowing of an oil-in-water precursor emulsion, pre-oxidation, and high-temperature calcination. A schematic diagram of the process and formation mechanism is shown in Figure 7. This ultra-fine silicon carbide has excellent thermal stability and good semiconductor properties, which also indicate its considerable application potential in wave absorption, electronic semiconductors, energy storage, and so on.

Because of the coverage of SiO_2_, the SiC fiber showed excellent performance compared to the oxide fiber and the nitride fiber, providing a new strategy to design the microstructures of fibers that can withstand high-temperature and oxidation conditions for long-term service.

### 2.4. Other Fibers

There are many types of ceramic fibers, and the above are only a few of the more widely used ceramic-based fibers. Other materials, such as titanium dioxide (TiO_2_) [56,57,58] and zirconium carbide (ZrC) [59], are also excellent ceramic-based fiber materials and have good prospects in various application fields.

## 3. Ceramic Aerogels

Aerogels are novel materials that have emerged in insulation research in recent years with nanoscale voids. A material becomes an aerogel when the gel is stripped of most of its solvent, or when the medium filling the gel space is a gas and the exterior is solid. Common methods for the synthesis of aerogels include the sol–gel method, sacrificial template method, solution spinning method, chemical vapor deposition method, etc. Figure 8 is a schematic diagram of another method of electrospinning to directly prepare ceramic aerogels [60]. However, there are two limitations in producing aerogels. One is the temperature, which is the reason that ceramic aerogels are very popular. The other is the inhibition of crystal transformation, which requires researchers to apply different treatments during the process. The main research directions of high-temperature aerogels are oxide aerogels, carbide aerogels, nitride aerogels, carbon aerogels, etc.

### 3.1. Oxide Aerogels

Common oxide aerogels are silicon oxide, aluminum oxide, and boron nitride. The classification depends on their chemical composition. Oxide aerogels can withstand very high temperatures and equally can exhibit defects such as inherent brittleness.

#### 3.1.1. Al_2_O_3_ Aerogel

The Al_2_O_3_ aerogel has a wide range of applications in thermal insulation, sound absorption, and catalysis due to its unique nanoscale three-dimensional network skeleton structure, ultra-low density, high porosity, and low thermal conductivity.

Zu [61] used the acetone–aniline in situ water generation method in combination to prepare an ultra-high-temperature-resistant strong alumina aerogel, with its brittleness greatly improved. The morphology and appearance of the formed alumina aerogel are shown in Figure 9. This alumina aerogel could not only withstand a high temperature of 1300 °C but also possessed improved mechanical strength. Sun [62] prepared composite high-temperature thermal insulation tiles using a mullite fiber-toughened Al_2_O_3_ aerogel, which enhanced the mechanical properties of the Al_2_O_3_ aerogel while maintaining good high-temperature thermal insulation properties.

In addition, doping with some additives (rare earth oxides, SiO_2_, and other oxides) can increase the sintering temperature of Al_2_O_3_ aerogels. Liu [63] prepared Al_2_O_3_-doped silica (SiO_2_) composite gels by the sol–gel method with a co-prepolymer, and, after freeze-drying and high-temperature heat treatment, Al_2_O_3_-SiO_2_ nanogels with a stable structure under high-temperature conditions were obtained. When the Al/Si molar ratio was 5:1, the aerogel maintained a large specific surface area (185 m^2^/g) and a uniformly distributed mesoporous structure (pore size of 6~8 nm) at 950 °C, whose infrared radiation (IR) reflectivity was as high as 40% in the wavelength range of 3~6 μm, effectively reducing the heat loss caused by IR thermal radiation. Zhou [64] prepared a Y_2_O_3_-Al_2_O_3_ aerogel by adding Al_2_O_3_·6H_2_O to YCl_3_·6H_2_O, and the specific surface area still reached 380–400 m^2^·g^−1^ after heat treatment at 1000 °C, indicating that the incorporation of Y_2_O_3_ improved the high-temperature resistance and thermal stability of Al_2_O_3_.

Electrostatic spinning technology can reduce the diameter of fibers to the nanometer level. The fibers obtained by this method possess excellent mechanical properties, electromagnetism, diverse chemical compositions, high specific areas, and other characteristics. Although the performance of alumina nanofibers is superior to that of alumina fibers, their mechanical properties still need to be improved. Li [18] doped SiO_2_ in the preparation of alumina nanofibers, and the structural surface of the fibers was dense, with no pores and cracks. Moreover, it possessed excellent flexibility and mechanical strength. Corrias [65] utilized a sol–gel process to produce nanocrystalline *γ*-Al_2_O_3_ and FeCo-Al_2_O_3_ nanocomposite aerogels constituted of FeCo alloy nanoparticles dispersed in the *γ*-Al_2_O_3_ nanocrystalline matrix. This indicated that calcination at elevated temperatures will give rise to *γ*-Al_2_O_3_ that is stable up to 1000 °C.

Zhang [66] synthesized a novel type of ultra-high-temperature insulating and strong ceramic nanorod aerogel (CNRA) by using alumina nanorods as the basic manufacturing unit and a small amount of silica solvent as the binder. The CNRAs obtained have superior properties, such as ultra-high heat resistance, low thermal conductivity, and high mechanical strength, as shown in Figure 10. Therefore, they have great advantages in extremely harsh environments. Mei [67] selected the nonionic surfactant polyethylene glycol as the pore size regulator of an alumina aerogel and regulated the pore size distribution of alumina by changing the molecular weight and additional amount of polyethylene glycol in the sol–gel process.

Ren [68] constructed a super-elastic silicon carbide aerogel using chemical vapor deposition and layer-by-layer self-assembly, as shown in Figure 11. This aerogel consists of a three-dimensional porous structure, and it possesses excellent particulate matter removal efficiency and high absorption capacity. Zou [69] synthesized alumina-based aerogels by acetone–aniline in situ aqueous formation, focusing on the effect of the modification temperature on the alumina-based aerogels. It was found that the morphology of the modified alumina-based aerogel will change from a network structure of needle-like particles to a larger sheet particle structure, and this structural transformation will cause the alumina-based aerogel to have better heat resistance. The thermal stability of the aerogel increases with the increase in the modification temperature. In addition, the modification can also inhibit the phase transition of alumina-based aerogels to α-Al_2_O_3_. Studies have also found that mullite felt and titanium dioxide can be added to alumina-based aerogels for further enhancement.

#### 3.1.2. ZrO_2_ Aerogel

Zirconia aerogel has drawn attention because of its high-temperature resistance up to 2200 °C. Meanwhile, researchers have found that doping with hybrid atoms or introducing nanofiber building blocks can further improve the performance.

Liu [70] prepared ZrO_2_ fibers with nano-diameters and abundant hydroxyl groups by using polyacetylacetonate zirconium as a precursor and compounded them with a ZrO_2_ aerogel to obtain high-performance fiber-composite ZrO_2_ aerogel heat mats, whose thermal conductivity was 0.026, 0.037, and 0.058 W·m^−1^·K^−1^ at 600, 800, and 1000 °C, respectively. Rosas [71] prepared ZrO_2_ nanofibers by electrospinning. For calcination at different temperatures, the surface of the fiber appears granular and exhibits a lower BET area with the increase in temperature. In addition, the temperature increase also leads to a lower steady-state conversion value for the ZrO_2_ fiber catalyst, where the semicrystalline tetragonal structure appears at 400 °C, and the positive monoclinic structure changes at 600 °C, but it leads to the growth of the monoclinic crystal size over 800 °C.

Chen [72] prepared a novel flexible zirconia nanofiber (ZNF) membrane by combining electrospinning and sol–gel methods, as shown in Figure 12. The flexibility and mechanical properties of ZNF membranes could be controlled by adjusting the grain phase and size of zirconia fibers. This flexible ZNF membrane has a high degree of filtration and excellent corrosion in acid and alkali solutions, which indicates its promising application prospects in the microfiltration of wastewater. Koo [73] reported a type of electrospun yttrium-stabilized zirconia (YSZ) nanofiber. The diameter of the electrospun YSZ nanofibers could be precisely controlled within the range of 200–900 nm by changing the viscosity of the precursor solution. The prepared YSZ had a wide, porous nanocrystalline structure with a high aspect ratio and high specific surface area.

#### 3.1.3. SiO_2_ Aerogel

The SiO_2_ aerogel [74,75] possesses very low thermal conductivity (0.01 W·m^−1^∙K^−1^) and excellent thermal insulation [76] properties, as a promising material to replace traditional insulation materials. However, silica aerogels exhibit inherent brittleness and poor mechanical strength and flexibility. Therefore, research has focused on resolving the abovementioned weaknesses [77]. It was found that a fibrillated aerogel could improve these defects, but fiber reinforcement will lead to the deterioration of its thermal insulation properties and density, and the compatibility of the fiber–silica interface represents a new problem that needs to be solved. Lee [78] prepared submicron-level composite fibers of SiO_2_/TiO_2_ with various components by calcination via sol–gel and electrospinning technology. It was noted that no gelling agents or binders were used in the preparation process, and the maximum amount of TiO_2_ for suitable fiber formation was around 50 moles. Moreover, the calcination temperature and TiO_2_ content greatly affected the surface morphology and crystallization behavior of the electrospinning fibers.

Tang [79] proposed the introduction of silica nanowires into the silica matrix as a secondary phase, which successfully solved this problem and synthesized a new silica nanowire–silica aerogel with excellent thermal insulation and mechanical properties. The data analysis clearly showed that the introduction of silica nanowires could improve both the performance and the mechanical properties of the composites with the increase in nanowires. Calisir [80] compared the nanofibrous silica mats produced via solution blowing and centrifugal spinning. It was found that samples fabricated by centrifugal spinning showed a more homogeneous and defect-free fibrous structure, with a higher average fiber diameter than in those obtained from solution-blowing. However, the solution blowing provided a more compact SiO_2_ fibrous mat with a smaller pore size, which was suited to filtration and separation applications. Moreover, the fluffier centrifugal spinning SiO2 outputs could be advantageously adopted for thermal insulation applications.

Wang [81] integrated flexible electroamplified silicon nanofibers with rubber-like Si-O-Si bonding networks to prepare biomimetic nanofiber (SNF) aerogels with superelasticity. The prepared SNF aerogel possessed ultra-low density (>0.25 mg·cm^−3^), high porosity, temperature-invariant superelasticity (up to 1100 °C), and strong fatigue resistance. In addition, it also had fire resistance and ultra-low thermal conductivity for the ceramic itself of 0.024 W·m^−1^∙K^−1^. It is a promising thermal insulation material, whose thermal insulation performance is shown in Figure 13, and it opens up additional possibilities for thermal insulation materials’ design and application.

Mi [82] reported a novel method for the preparation of three-dimensional silica fiber sponges. In the course of the study, it was found that three-dimensional fibers were easily collected on the aluminum foil using a properly aged tetraethyl orthosilicate (TEOS)/PVA solution. The prepared silica sponge has a continuous fiber structure, low bulk density (16 mg·cm^−3^), high surface area (6.5 m^2^·g^−1^), heat resistance, etc. Because of these excellent properties, the sponge can be used in various fields. Huang [83] proposed a silica nano-aerogel (SNFA) prepared by using electrospun silica nanofibers as a matrix and silica sol as a high-temperature nano-glue. This method addresses the disadvantage of the inherent brittleness of aerogels. The prepared silica aerogel exhibits low density, high-temperature resistance from flexibility to rigidity, adjustable mechanical properties, and so on.

Tepekiran [84] prepared silica-based nanofibers via centrifugal spinning and subsequent calcination. It was indicated that the 15 wt%TEOS/PVP sample possessed the highest flexibility and filtration efficacy among all the silica nanofibers. The average fiber diameter of the optimized web was found to be the lowest, around 521  ±  308 nm, which resulted in enhanced filtration efficiency of around 75.89%.

Similar to oxide fibers, oxide aerogels also exhibit potential for applications in high-temperature oxidized conditions, especially due to their excellent thermal insulation performance. However, their shortcomings in the grain-coarsening process over 1300 °C still limit their utilization. Therefore, the goal has not changed as researchers are still searching for more efficient materials for higher-temperature utilization, such as aerogels of carbide or nitride.

### 3.2. Nitride Aerogels

Nitrides, with good thermal stability, chemical inertness, and a low coefficient of thermal expansion, have been used to synthesize aerogels with excellent high-temperature thermal and mechanical stability.

#### 3.2.1. BN Aerogels

BN has a structure similar to graphite crystals and has four different variants: hexagonal boron nitride (*h*-BN), rhombic boron nitride (*r*-BN), cubic boron nitride (*c*-BN), and fibrillated boron nitride (*w*-BN). BN fibers themselves have a low dielectric constant, high radiation absorption, high thermal conductivity, good mechanical properties, etc., and can be widely used in optical, thermal, and electrical applications, and so on. Moreover, porous boron nitride nanomaterials have the advantages of low density, a large specific surface area, superior mechanical properties, and good chemical stability. Boron nitride (BN) aerogel is a three-dimensional nanoporous material with a high specific surface area, large porosity, and low density. At the same time, it inherits good insulation, oxidation resistance, thermal stability, and chemical stability from conventional BN.

When combined with other different forms of aerogels, *h*-BN can improve the thermal and mechanical properties. Wang [85] prepared hexagonal boron nitride (*h*-BN)/TEMPO-oxidized nanocellulose (TOCNF) aerogels with different pore structures, and then compounded TOCNF to prepare *h*-BN/TOCNF composite films. The results showed that when the solid content ratio of *h*-BN to TOCNF was 3:1, TOCNF could play a good role in dispersing *h*-BN, and the thermal conductivity of the composite film was the most efficient. The strength of the composite film was the best when the solid content ratio of *h*-BN and TOCNF was 1:1, the elongation at break was approximately 13%, and the tensile strength was 24.8 MPa, indicating its excellent mechanical properties.

It is well known that ceramic aerogels possess poor thermal stability and will degrade under thermal shock. On this basis, Xu [86] synthesized hexagonal boron nitride aerogels (hBNAGs) using a specially designed 3D graphene aerogel membrane plate. Its characterization is shown in Figure 14. The hBNAG has a double-paned metastructure hyperbolic structure with a negative Poisson’s ratio (−0.25) and a negative thermal expansion coefficient (−1.8 × 10^−6^ per °C). The hBNAGs synthesized by this method exhibit ultra-low density (0.1 mg^−1^·cm^3^), superelasticity (up to 95%), thermal superinsulation (0.0024 W·m^−1^·K^−1^ in vacuum, and 0.0020 W·m^−1^·K^−1^ in air), and thermal stability under sharp thermal shocks (~275 °C/s) and long-term high-temperature exposure (900 °C in air and 1400 °C in vacuum).

Lu [87] proposed an effective “growth-space assisted” method for the large-scale production of elastic silicon carbide nanowire aerogels (SiC NWAGs) with controllable density and shape, in view of the shortcomings of the complex and low-yield synthesis process of elastic ceramic aerogels. The SiC NWAGs obtained by this method show excellent deformability, processability, high-temperature stability, and fire resistance, etc., addressing the shortcomings of the high brittleness and poor high-temperature stability of traditional aerogels very well. This also expands the applications of aerogels in harsh environments and complex compositions.

Yang [88] prepared BN/SiOC aerogels by the sol–gel method, and the mechanical properties of SiOC aerogels were improved by adding BN particles to increase the density of the composite. The aerogel exhibited high compressive strength and high heat resistance at 1300 °C. Aravind [89] reported a porous silicon carbide oxide glass synthesized with a highly porous ambiguity material. The synthesized ambiguous gel had a very high surface area and pore volume, and the study found that the porosity remained unchanged during the pyrolysis of its ambiguous gel in an inert gas. Zeng [90] reported a simple cryocasting process (a method for the controlled freezing of an inorganic suspension in water) to prepare a boron nitride nanosheet (BNNS) aerogel, which had an ordered and anisotropic microstructure, as well as anisotropic superelasticity, high compressive strength, and great absorption capacity. It has wide application prospects in the fields of catalysts, environmental remediation, and energy absorption.

Li [91] developed a novel boron nitride nanoribbon aerogel by the high-temperature amination reaction of a melamine diborate precursor, as shown in Figure 15. Its extremely unique structure provides it with excellent temperature invariance, outstanding compressive elasticity, bending elasticity, torsional elasticity, cutting resistance, and recovery ability. In the range from liquid nitrogen temperatures (−196 °C) to over 1000 °C, these BN nanoribbon aerogels maintain superior mechanical superflexibility.

#### 3.2.2. Si_3_N_4_ Aerogels

Silicon nitride (Si_3_N_4_)-based composites, with good wave transparency and acceptable high-temperature mechanical properties, are considered excellent candidates for aircraft radomes.

Kong [92] prepared Si_3_N_4_ aerogels by the carbothermal reduction method with flowing nitrogen in order to break through the limitation of silica aerogels to 600 °C and to improve their stability. Ultra-light α-Si_3_N_4_ aerogels with controllable density can be prepared by a simple heat treatment process. They have excellent properties, such as fire resistance and elastic compressibility at 1200 °C, being a promising material for thermal insulation applications. Su [93] prepared ultra-light α-Si_3_N_4_ nano-felt aerogels (NBAs) with tunable density (1.8–9.6 mg·cm^−3^). α-Si_3_N_4_ NBA has ultra-high elastic compressibility, fire resistance, thermal insulation, and an ultra-low dielectric constant, as exhibited in Figure 16. With the use of silica sol and carbon black as raw materials, Zhang [52] synthesized Si_3_N_4_ nanowire woven microspheres via a simple and efficient method. The Si_3_N_4_ nanowire microspheres formed by this method have a uniformly curved morphology and high specific surface area and are very stable in a gas–solid growth environment.

Compared to oxide aerogel, nitride aerogel also displayed similarly higher utilization potential as nitride fibers in high-temperature conditions up to 1800 °C. However, there still exist the disadvantages of worse oxidation resistance and a lack of suitable utilization in high-temperature oxidation conditions for long periods. Therefore, it is necessary to choose a non-oxide aerogel with oxidation resistance for higher-temperature utilization, such as a carbide aerogel.

### 3.3. Carbide Aerogels

Carbide itself has excellent high-temperature resistance, wear resistance, corrosion resistance, a high melting point, high hardness, good electrical conductivity, and good mechanical properties. Carbide aerogel is also the most promising material to resist temperatures above 1200 °C and is widely used in the aerospace field. Carbide aerogels have higher temperature resistance compared to traditional oxide aerogels, and their types are also very abundant. The preparation of carbide aerogels is complicated by the oxidation reaction of carbon, commonly known as SiC aerogels.

#### SiC Aerogel

Silicon carbide aerogel is the most popular carbide aerogel, which is usually obtained by mixing silica aerogel with carbon in a reduction reaction. SiC aerogels have excellent high-temperature resistance and stability and can be successfully applied above 1300 °C. The chemical and high-temperature stability and multifunctional properties of SiC aerogels indicate their potential application in electromagnetic absorption and thermal insulation in harsh environments.

Mahalingam [94] used a facile method consisting of simultaneous centrifugal spinning and solution blowing to mass-produce SiOC fibers from preceramic polymers. It was indicated that the fiber diameter and morphology were influenced by the rotating speed, working pressure, and polymer concentration, where non-porous fibers could be obtained by using a low-volatility binary solvent system and after pyrolysis at a high temperature. This fabrication method, based on pressurized gyration, showed great promise in the mass production of ceramic fiber bundles.

Liang [95] used an eggplant-derived method to fabricate a green and convenient SiC aerogel by using the naturally porous structure of eggplant decorated with in-situ-grown SiC nanowires. The SiC aerogel thus obtained has low density (Figure 17a), mechanical stability, thermal stability, and thermal insulation properties (Figure 17b). Su [96] successfully prepared silicon carbide aerogels with entangled nanowires and highly porous structures by nucleating silicon carbide nanowires through a chemical deposition reaction on graphite caps.

The porosity of single-porous silicon carbide is generally considered to be high at approximately 30%V/V, and its preparation process is usually more complicated. On this basis, Leventis [97] crosslinked the 3D sol–silicon nanostructure prepared by carbothermal reduction with polyacrylonitrile (PAN) synthesized by a simple mixed monomer to obtain silicon carbide with porosity of 70%V/V. The morphology of the silicon carbide network is constant at the processing temperature of 1300~1600 °C. Samples processed at 1200 °C are mesoporous and amorphous.

Due to their excellent heat resistance up to a maximum operating temperature of 1200 °C, oxidation resistance, and high strength, SiC fibers are also employed as high-temperature-resistant and reinforcing materials for a wide range of promising applications. Chen [98] experimentally showed that adjustable three-dimensional porous silicon carbide nanowire scaffolds can be prepared by adjusting the solid loading content in the mesh melamine foam template. These SiC nanowire scaffolds not only have high strength and good fire-stopping performance but also could efficiently absorb oil or act as catalysts, etc.

Song [99] prepared a multifunctional silicon carbide nanofiber aerogel (SiC NFAS) spring using a simple thermochemical process, as shown in Figure 18. This SiC NFAS has ultra-low density, ultra-low thermal conductivity, as well as great mechanical properties, showing strong superelasticity and cyclic fatigue resistance at temperatures ranging from low to high.

Owing to the surface oxidation of SiC, the covered SiO_2_ protected the inner SiC from further oxidation, which showed the excellent performance of the SiC aerogel compared to the oxide aerogel or that of nitride, and provided a new approach to design the microstructure of carbide aerogel to improve its oxidation resistance under high-temperature oxidation conditions, for future utilization in long-term service.

On basis of the above-introduced ceramic-based materials, both the nanofibers and aerogels show their application potential in high-temperature harsh conditions. A comparison of the material and preparation methods of the different nanofibers and aerogels is shown in Table 2, which offers a reference for researchers regarding the main flexible nanomaterials with a series of materials, their fabrication processes, as well as their properties. Because of properties such as low density, porosity, a large surface area, low thermal conductivity, high-temperature resistance, and chemical stability, they show potential for the application fields of thermal insulation, air filtration, water treatment, sound absorption, electromagnetic wave insulation, battery separators, and catalytic applications.

## 4. Applications

### 4.1. Thermal Insulation

Due to their low density, low thermal conductivity, and chemical stability, ceramic fibers have a wide range of applications in the field of thermal insulation [113], especially in aerospace and other fields. Generally, the lower the density, the lower the thermal conductivity and the better the thermal insulation performance of the material will be.

Si [103] combined silica nanofibers with an aluminum–borosilicate matrix to prepare ceramic nanofiber aerogels (CNFAs) with a layered structure and excellent elasticity. CNFAs exhibit good mechanical properties over a wide temperature range, as shown in Figure 19. The density and shape of such CNFAs is adjustable, showing the comprehensive characteristics of fly weight density >0.15 mg·cm^−3^, and they could be stabilized at 1100 °C to provide outstanding fire resistance and thermal insulation.

Jia [114] prepared a SiO_2_-Al_2_O_3_ composite ceramic sponge with temperature invariance and high compressibility in a simple manner, leading to an advanced anisotropic layered ceramic sponge with a density as low as 10 mg·cm^−3^, thermal conductivity as low as 0.034 W·m^−1^∙K^−1^, and temperature-invariant compressive elasticity in the temperature range of −196~1000 °C, as shown in Figure 20. The various excellent parameters make these ceramic sponges promising in terms of thermal insulation materials. Liu [54] prepared hollow silicon carbide fiber pads doped with nitrogen using sacrificial stencils and electrospinning. The cavity walls of the hollow fibers are thin, with excellent performance in terms of flexibility, acid and alkali resistance, non-combustibility, and high-temperature stability in air or inert gases. This material has thermal conductivity of 0.026 ± 0.013 W·m^−1^∙K^−1^ and can be used as a high-temperature heat insulator. Su [96] reported a highly porous SiC nanowire aerogel (NWA) that improved the brittleness of traditional aerogels and the weakness of volume shrinkage at high temperatures. These SiC NWAs are composed of many interwoven nanowires, with extremely low density, large recoverable compressive strain, fire resistance, and high-temperature and oxidation resistance. These properties make it a promising fire-resistant and high-temperature insulation material.

### 4.2. Air Filtration

Air pollution is a serious problem worldwide, especially in developing countries, where the pollution is mainly derived from the particulate matter (PM) produced by cars, power plants, chemicals, etc. PM is also known as lung particle matter. It remains in the atmosphere for a long time, spreads across large distances, and contains a large number of harmful substances. In addition, it can easily remain in the bronchi and alveoli, and even enters the blood, which is extremely harmful to health. The effective removal of PM has become an urgent need for contemporary development. Therefore, an efficient filtration material is necessary to address this situation.

Jia [115] prepared an Al_2_O_3_-stabilized ZrO_2_ (ASZ) submicron fiber air filter paper with excellent flexibility and stability for use in solution-blowing and calcination technology. The ASZ paper, with an area density of 56 mg·cm^−2^, has filtration efficiency of up to 99.56% at an airflow velocity of 5.4 cm·s^−1^ for 15~615 nm NaCl particles. Furthermore, this foldable all-ceramic air filter material could be an important solution for particle removal in high-temperature exhaust gases.

Wang [47] reported a porous three-dimensional sponge composed of yttrium-stabilized ZrO_2_ (YSZ) nanofibers with a density of 20 mg·cm^−3^ and a light weight. The elasticity of this sponge can be maintained at room temperature and high temperatures, and the filtration efficiency can reach 99.4% at room temperature and 99.97% at high temperatures, as shown in Figure 21. The YSZ nanosponge, therefore, has promising application prospects, and a practical automotive exhaust filter with filtration efficiency of 98.3% has been assembled.

Li [116] reported a typical structural model of a high-efficiency particulate air filter based on carbon nanotubes. High filtration efficiency can be obtained due to the aggressively high surface of the carbon nanotubes, and a low-pressure drop can be obtained by controlling the thickness and SVF value of the carbon nanofilm. Gradient nanostructures or layered nanostructures not only have higher filtration efficiency, but also have a longer service life and low pressure drop. Zhang [117] developed a novel hydroxyapatite (HAP) nanowire-based inorganic aerogel, shown in Figure 22, which can also be used as a high-efficiency air filter for PM2.5, in addition to its advantages of high porosity, ultra-light density, and low thermal conductivity. Compared with organic aerogels, HAP nanowire aerogels have the advantages of environmental protection, low costs, and biocompatibility.

### 4.3. Water Treatment

Water pollution is also a very serious problem, caused by the discharge of various types of sewage, oil spills, etc. For this reason, it is necessary to develop a system that can absorb and remove organic pollutants from polluted water. Many flexible ceramic fibers can be used as new materials to remove various organic pollutants.

Su [96] reported a highly porous three-dimensional silicon carbide nanowire aerogel (NWA) with recoverable compressibility and excellent high-temperature stability. The SiC NWA has high absorption capacity due to its high porosity and low density, as shown in Figure 23. The hydrophobic SiC NWA can selectively adsorb low-viscosity organic solvents. Moreover, this SiC NWA has elasticity and fire resistance, and could be used to absorb all organic solvents by squeezing NWA and burning it in air for heating. In addition, it was found that extrusion reabsorption does not lead to a decrease in absorption capacity, proving the SiC NWA as a promising material in sewage treatment and medicine cabinets. Ren [68] prepared SiC nanofiber aerogels (SCS-NAs) with high hydrophobicity using octadecyl trichlorosilane (OTS) as a modifier, which can also absorb oily pollutants and organic solvents by extrusion and combustion, with potential application value in remedying chemical leakages and oil leakages.

BN aerogels are also highly hydrophobic and can adsorb up to 160 times their weight in oil [118]. Xue [119] developed a multifunctional foam for cell networks composed of an interconnected nanotube hexagonal BN (*h*-BN) structure, which can well adsorb various hydrophobic oils in oil-contaminated water, with excellent antiseptic properties against strong acids and alkalis. This demonstrates that the three-dimensional tubular BN cellular-network foam (3D-BNF) is a very useful material for cleaning in very harsh environments. According to research statistics, most of the non-oxide ceramic materials are used for water treatment, such as SiC and BN, as mentioned above. For this reason, the application of oxide ceramic fiber as an absorbent material has yet to be studied [6].

### 4.4. Sound Absorption

Noise pollution has become indispensable in modern life with the rapid development of construction and transportation. It causes great inconvenience and impacts our lives and work. It is therefore crucial to develop materials that can absorb and block noise pollution. Here, ceramic materials can be used as an efficient and suitable sound-absorbing material.

Jia [114] prepared SiO_2_-Al_2_O_3_ composite ceramic sponges by a simple process. The layered structure, rough microfiber surface, and fiber vibrations of the SAC foam provide it with excellent sound absorption properties, as shown in Figure 24.

### 4.5. Electromagnetic Wave Absorption

With the rapid development of modern technology, electromagnetic pollution is becoming more and more serious, so it is of great significance to develop a material that can absorb electromagnetic waves. According to research, the most common electromagnetic-wave-absorbing material is SiC [120,121] fiber-based material, with an adjustable microstructure and dielectric properties.

Cai [122] prepared a series of highly porous, well-interconnected SiC@C nanowire foams (SCNFs) via a simple glucose solution permeation–carbonization method using silicon carbide nanowire aerogels as raw materials, which improved the shortcomings of the narrow absorption bandwidth and low absorption intensity of electromagnetic waves (EMWs). This SCNF has a density of 108 mg·cm^−3^, covering the entire absorption bandwidth of the X and Ku bands with an intensity of −52.5 dB. Moreover, it is advantageous for applications in extreme environments due to its great hydrophobicity and self-cleaning ability, as a promising new generation of EMW-absorbing materials.

Hou [123] prepared silicon carbide nanofiber pads with and without fiber arrangement by electrospinning and found that the arrangement of nanofibers can further improve the microwave absorption performance, as shown in Figure 25. In addition, the low weight fraction of SiC nanofiber pad silicone resin composites makes them a low-cost absorber. In addition, Hou [124] found that introducing hafnium carbide (HfC) into SiC nanofiber pads can not only improve their flexibility but also improves the dielectric properties and microwave absorption properties.

### 4.6. Battery Separators

Batteries are essential in our lives, among which Li-ion batteries provide efficient power for humans in many applications. However, they easily catch fire due to short circuits, which are mostly related to polymer separators, and they generally degrade immediately under sudden high temperatures. For this reason, it is necessary to develop ceramic fiber separators that withstand high temperatures.

Yan [106] prepared a series of oxide ceramic nanofiber films by sol–gel electrospinning technology, and the separator prepared by this film showed greater than 900% electrolyte absorption and high thermal insulation performance, which can dramatically improve the safety of lithium batteries. Zhao [125] prepared an elastic solid electrolyte to provide fast and continuous ion transport channels by filling the well-arranged Li_6.4_La_3_Zr_2_Al_0.2_O_12_(LLZO) nanofiber film with a sol–gel electrospinning method, which showed high ionic conductivity of up to 1.16 × 10^−4^ at 30 °C. The elastic surface exhibited stable Li plating/stripping cycling over 700 h, the maximum in the present research, as shown in Figure 26. Therefore, most of the present research on ceramic fiber battery separators mainly focuses on LLZO materials. However, studies on the application of other materials in battery separators are also carried out to obtain safer and more successful chemical ceramic fiber separators.

### 4.7. Catalytic Application

Ceramic fibers have high porosity and a beaten surface area and can be an attractive material for catalysts. Ceramic fibers have two forms in the field of catalysis: one is that the fiber itself has excellent catalytic activity for processing into ceramic flakes for use in the catalytic field, and the other is as a catalyst carrier.

Zhang [126] synthesized a composite nanofiber cathode composed of CeO_2_ particles and La_0.6_Sr_0.4_Co_0.2_Fe_0.8_O3−δ (LSCF) by coaxial electrospinning, which had significant ORR activity and durability. At 700 °C, the polarization resistance of this cathode is approximately one fifth of that of LSCF powder. This LSCF/CeO_2_ composite cathode possesses stability in anode separation cells.

When ceramic fibers are used as catalyst carriers, there are two methods by which to prepare the materials. One is to add a catalyst or catalyst precursor directly to the spinning solution, and the other is to load the catalyst onto the surface of the ceramic fiber after spinning calcination. Wang [127] synthesized Cu-Al_2_O_3_ fiber membranes by electrospinning technology, which exhibited high Fenton catalytic activity under neutral conditions. Cheng [128] synthesized ZnO/γ-Al_2_O_3_ nanofibers by electrospinning and calcination. Silver nanofibers are modified on the surfaces of nanofibers by reduction, and Ag/ZnO/γ-Al_2_O_3_ nanofibers exhibit the efficient catalytic degradation of methyl orange under ultraviolet irradiation.

## 5. Summary and Prospects

Developments and applications of ceramic-based nanomaterials such as nanofiber materials and aerogels have achieved tremendous progress in the past few years, including several types of oxides, such as Al_2_O_3_, ZrO_2_, and glass; carbide for SiC; nitride for BN and Si_3_N_4_; and other ceramic-based nanofibers. We, therefore, have presented a summary of these ceramic-based nanomaterials in terms of their types and their applications in the fields of thermal insulation, air filtration, water treatment, sound absorption, electromagnetic wave insulation, battery separators, catalysis, etc. Among them, we focused on summarizing the applications of ceramic-based nanomaterials for thermal insulation. In addition, in terms of improving the application of ceramic-based nanomaterials for thermal insulation, especially for high-temperature thermal insulation, we offer several pieces of advice on the development directions of ceramic-based nanofibers.

(1) To improve the efficiency of the thermal insulation of ceramic-based materials, ultra-fine nanofibers are essential. Therefore, it is necessary to employ methods such as centrifugal spinning, electrospinning, solution blow spinning, and self-assembly for the preparation of nanofibers.

(2) Polymers are usually used as thickeners in the spinning solution, which is removed after calcination and results in a fiber diameter reduction, porous surface formation, and decreased mechanical capacity. This phenomenon is eliminated by improving the solution by enhancing the solid content ratio as much as possible and avoiding polymer utilization.

(3) For some single-component ceramic-based nanofibers, their performance in terms of high-temperature resistance will decrease because of the nanometer effect. It is, therefore, necessary to add some other components to improve them, such as adding some yttrium oxide to improve the performance of α-Al_2_O_3_ nanofibers.

(4) The required quantities of nanofiber materials in actual applications for thermal insulation and sound absorption are huge, but most of the spinning methods’ efficiency meets the demands. Among all sol–gel routes, the solution blow spinning method possesses the highest efficiency for the preparation of nanofiber products as films and sponges, which should be given more research attention.

## Figures and Tables

**Figure 1 gels-09-00208-f001:**
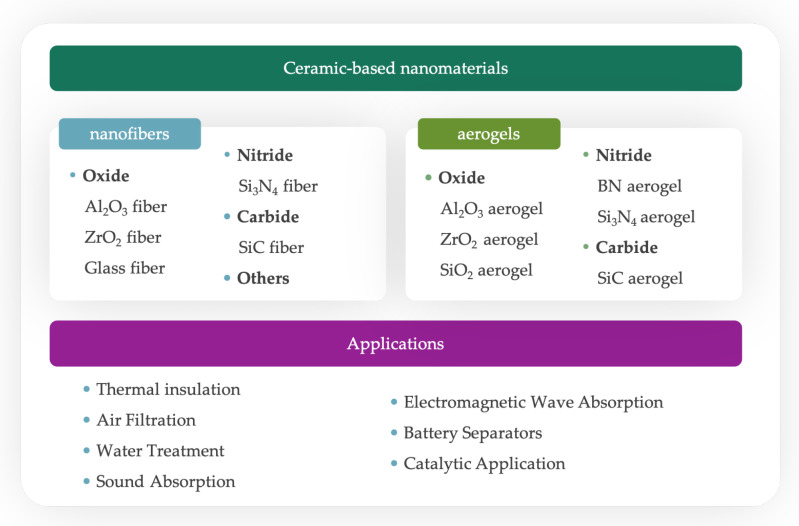
Research progress of ceramic-based nanofibers in terms of materials and applications.

**Figure 2 gels-09-00208-f002:**
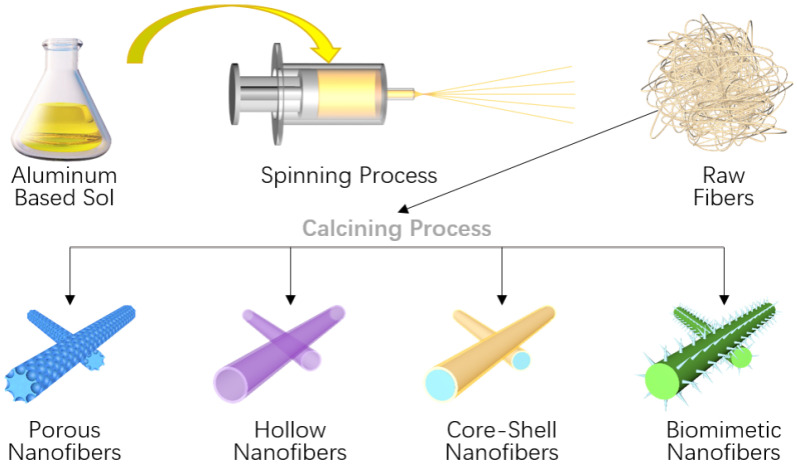
Structure construction for sol–gel ultra-fine alumina fiber (adapted with permission from Ref. [29]. 2020, Elsevier).

**Figure 3 gels-09-00208-f003:**
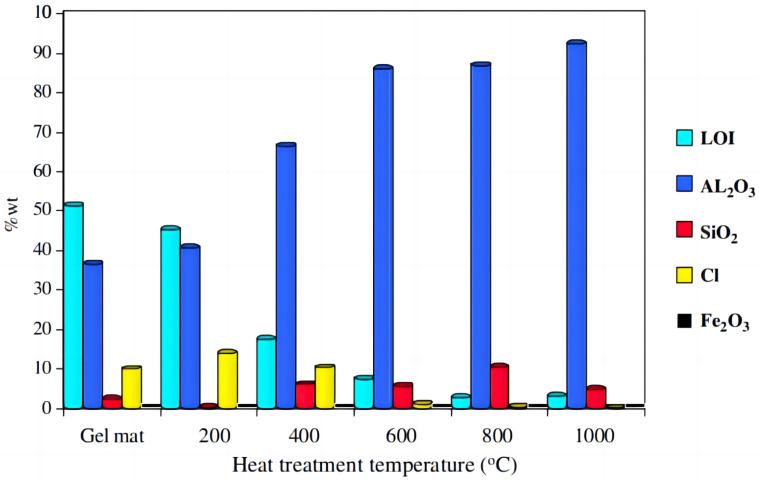
Chemical composition of the alumina mat (4 wt% SiO_2_) at different temperatures with a holding time of 4 h by XRF analysis after the loss of ignition (L.O.I.) test (adapted with permission from Ref. [32]. 2006, Elsevier).

**Figure 4 gels-09-00208-f004:**
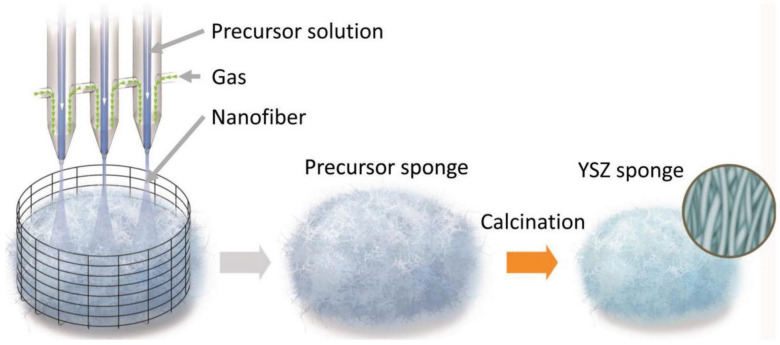
Fabrication process of YSZ nanofiber sponge (adapted with permission from Ref. [47]. 2018, WILEY-VCH Verlag GmbH Co. KGaA).

**Figure 5 gels-09-00208-f005:**
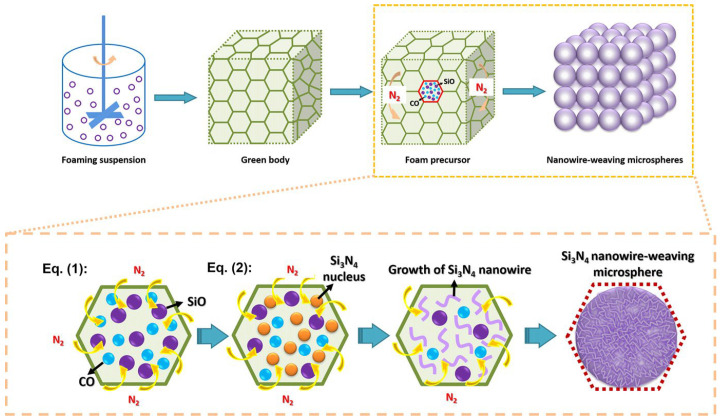
Schematic diagram of the formation of nanowire-weaving microspheres (adapted with permission from Ref. [52]. 2019, The American Ceramic Society).

**Figure 6 gels-09-00208-f006:**
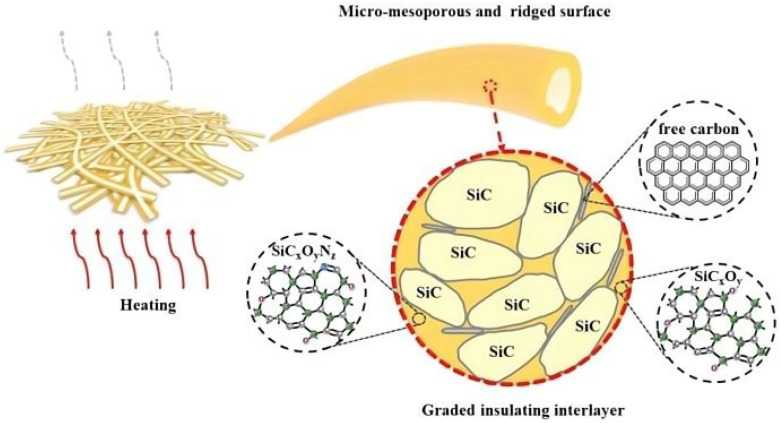
Schematic illustration of integration of specific multilayer scattering mechanisms in the as-obtained N-doped hollow SiC fibers (adapted with permission from Ref. [54]. 2017, Royal Society of Chemistry).

**Figure 7 gels-09-00208-f007:**
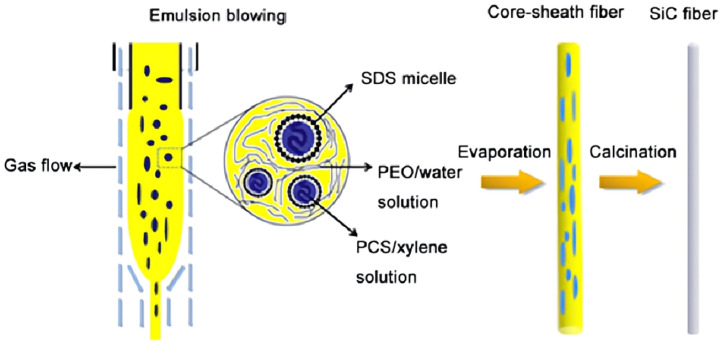
Schematic of overall procedure and the formation mechanism of SiC fibers via emulsion–blow spinning (adapted with permission from Ref. [55]. 2017, Transactions of the Indian Ceramic Society).

**Figure 8 gels-09-00208-f008:**
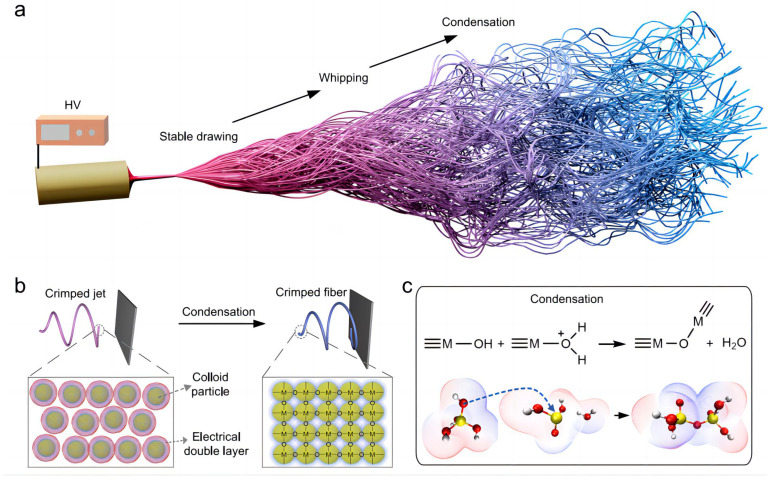
Illustration of 3D reaction electrospinning for direct fabrication of ceramic nanofibrous aerogels. (**a**) shows the 3D reaction electrospinning process. (**b**) shows the highly reactive colloidal particles forming highly crosslinked and robust skeletons via condensation and jet solidification. (**c**) shows the distance between the protonated oxygen atom of ≡M-OH and the metal atom increase, creating a good leaving group (adapted with permission from Ref. [60]. 2022, The Authors).

**Figure 9 gels-09-00208-f009:**
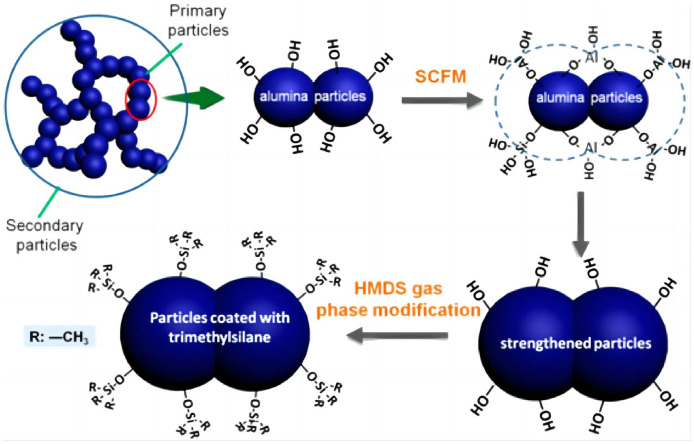
Morphology and appearance of alumina aerogel (adapted with permission from Ref. [61]. 2013, American Chemical Society).

**Figure 10 gels-09-00208-f010:**
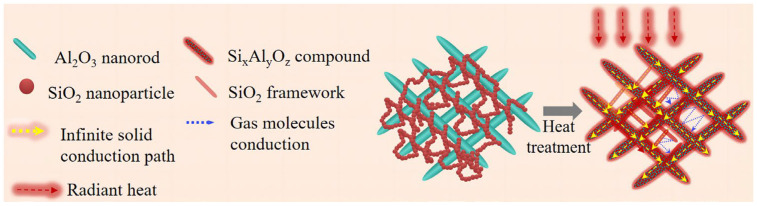
Schematic illustration of the structural strength and thermal insulation performance of CNRAs (adapted with permission from Ref. [66]. 2021, American Chemical Society).

**Figure 11 gels-09-00208-f011:**
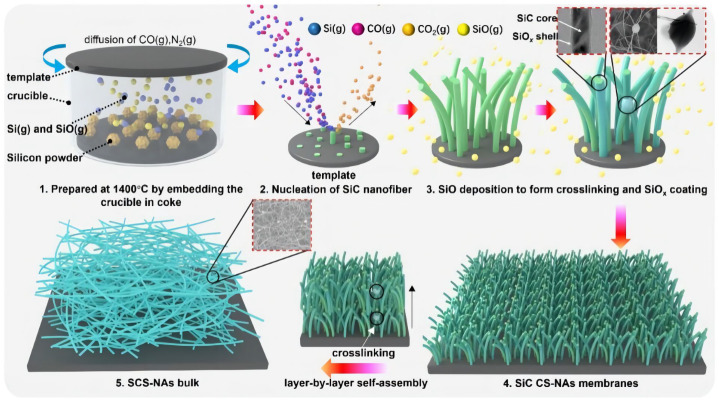
Suggested growth mechanism of SCS-NAs bulk-assembled by crosslinked SiC/SiOx core–shell nanofibers (adapted with permission from Ref. [68]. 2019, American Chemical Society).

**Figure 12 gels-09-00208-f012:**
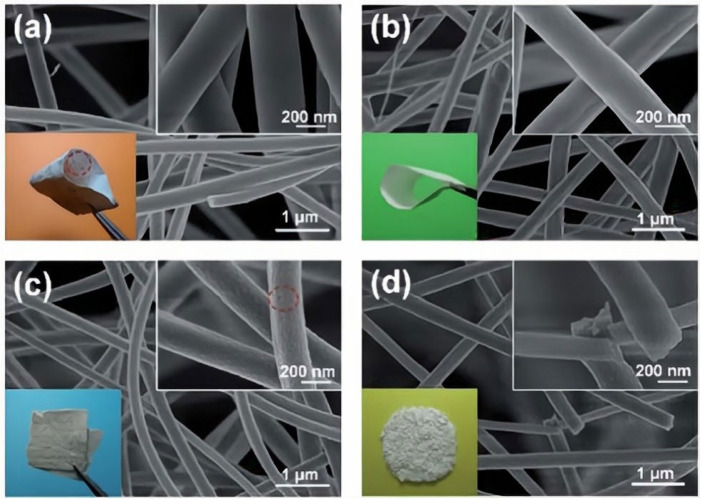
FE-SEM images of (**a**) ZNF1@600, (**b**) ZNF1@800, (**c**) ZNF1@1000, and (**d**) ZNF0 membranes. The insets are the optical images of the corresponding membranes (adapted with permission from Ref. [72]. 2014, Royal Society of Chemistry).

**Figure 13 gels-09-00208-f013:**
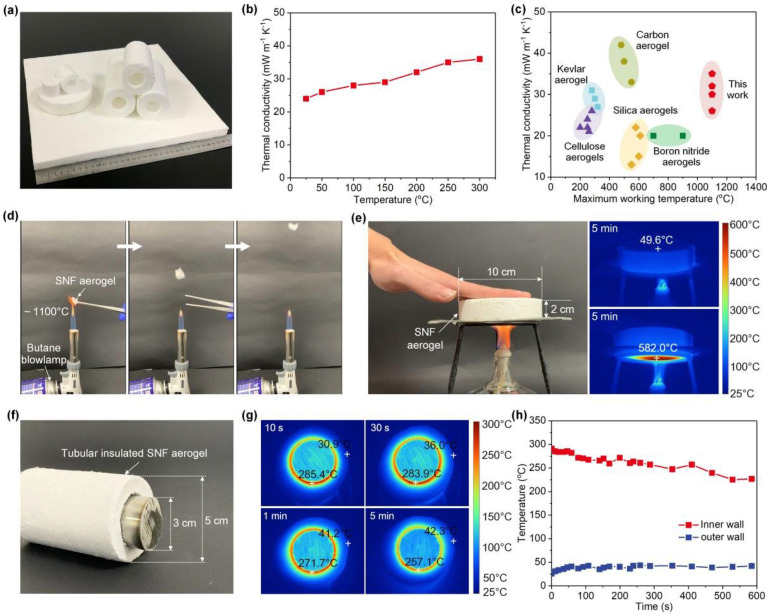
(**a**) Optical image of large-sized SNF aerogels in different shapes for thermal insulation applications. (**b**) Thermal conductivity of SNF aerogels versus testing temperature. (**c**) Comparison of the thermal conductivity and maximum working temperatures of SNF aerogels and other reported cellular materials. (**d**) Ultralight and fire-resistant qualities of an SNF aerogel. (**e**) Thermal insulation capacity of an SNF aerogel. (**f**) Optical image of a tubular insulated SNF aerogel. (**g**) Infrared images of the thermal insulation process of a tubular SNF aerogel. (**h**) The time-dependent temperature profile of the inner and outer walls of the aerogel (adapted with permission from Ref. [81]. 2020, WILEY-VCH Verlag GmbH Co. KGaA).

**Figure 14 gels-09-00208-f014:**
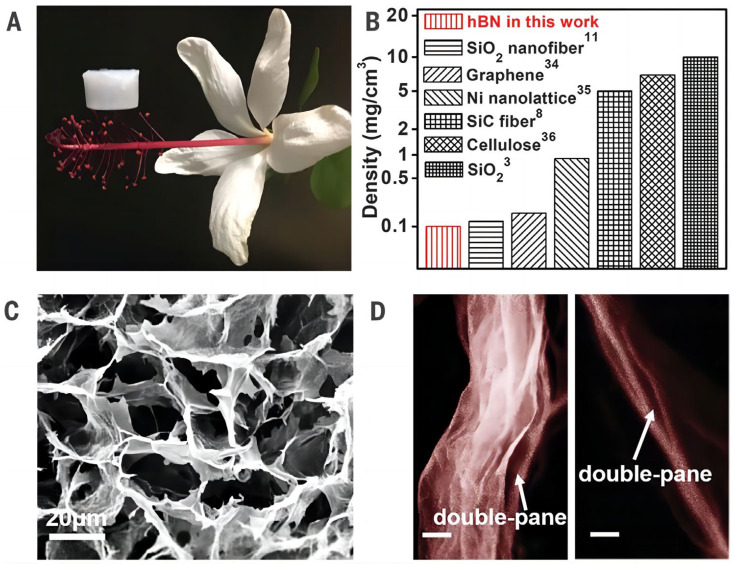
Material characterization of *h*-BN aerogels. (**A**) An optical image showing an *h*-BN aerogel sample resting on the stamen of a flower. All tests were performed on ceramic aerogels with a density of 5 mg·cm^−3^, unless otherwise noted. (**B**) The lightest *h*-BN aerogel sample compared with other ultralight materials. The superscript numbers indicate the corresponding referenced work. (**C**) SEM image of *h*-BN aerogel. (**D**) SEM images of the double-pane wall structure of *h*-BN aerogel. Scale bars, 20 nm (adapted with permission from Ref. [86]. 2019, The American Association for the Advancement of Science).

**Figure 15 gels-09-00208-f015:**
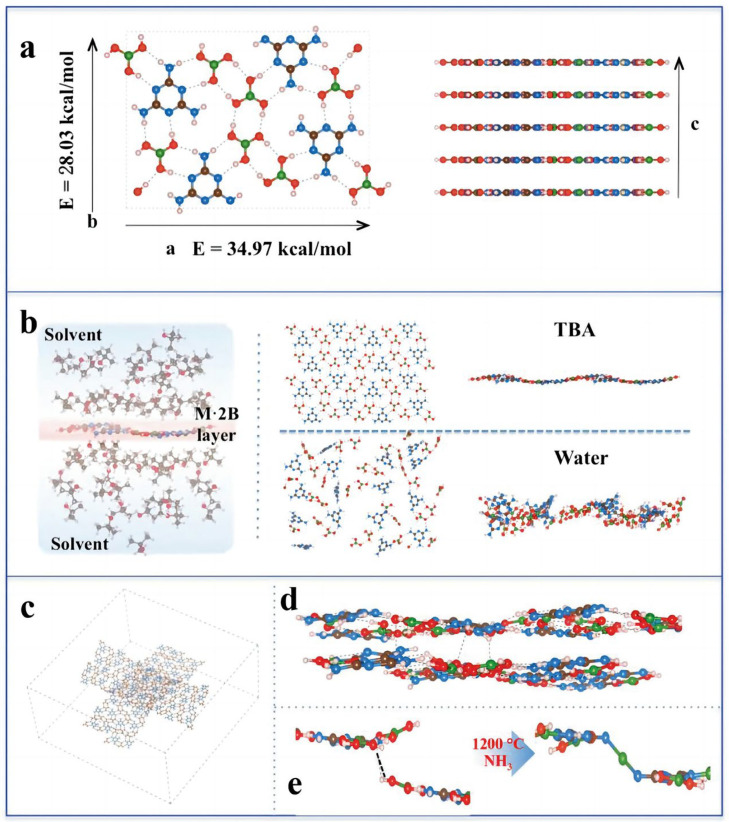
(**a**) The atomic structure of an M·2B cell in top (**left**) and side (**right**) view. The hydrogen bonds are presented with dashed lines. (**b**) Simulation setup of a single layer of M·2B embedded in solvent (left), snapshots of M·2B after equilibration in pure TBA (top) and pure water (bottom) (right). (**c**) Simulation of two crossed M·2B layers. (**d**) The side view of two crossed M·2B layers, linked to each other via the H-bond at the edge of the layer. (**e**) Simulated illustration of the H-bond converted to a BN bond at the edge of the layer (boric acid (B) and melamine (M)) (adapted with permission from Ref. [91]. 2019, WILEY-VCH Verlag GmbH Co. KGaA).

**Figure 16 gels-09-00208-f016:**
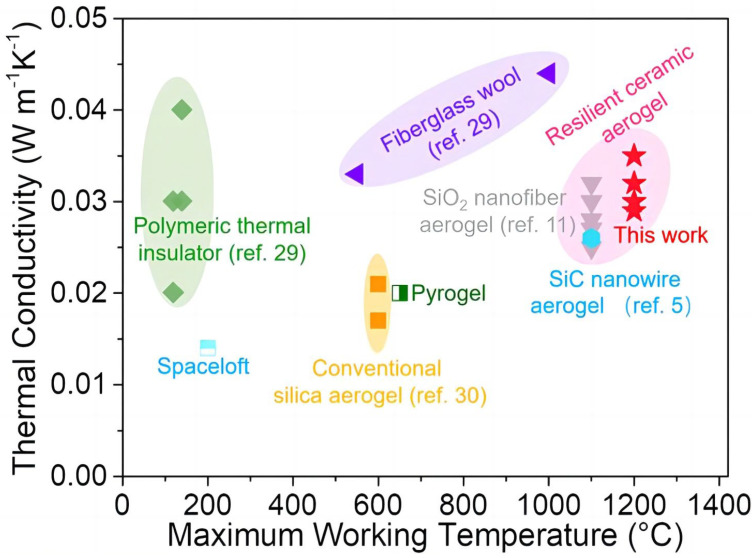
Thermal conductivity versus the maximum working temperature of thermally insulating materials (adapted with permission from Ref. [93]. 2019, American Chemical Society).

**Figure 17 gels-09-00208-f017:**
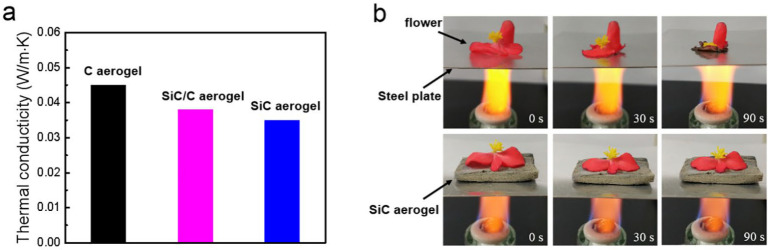
(**a**) Thermal conductivity of carbon, SiC/C, and SiC aerogels and (**b**) heat resistance properties of the SiC aerogels (adapted with permission from Ref. [95]. 2019, Elsevier).

**Figure 18 gels-09-00208-f018:**
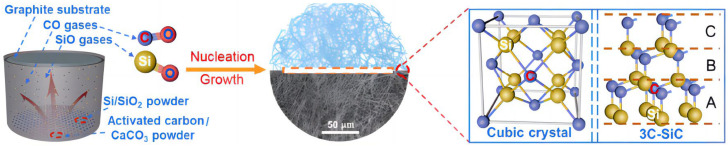
Schematic illustration of SiC NFAS fabrication and corresponding atomic structure model (adapted with permission from Ref. [99]. 2022, The Authors).

**Figure 19 gels-09-00208-f019:**
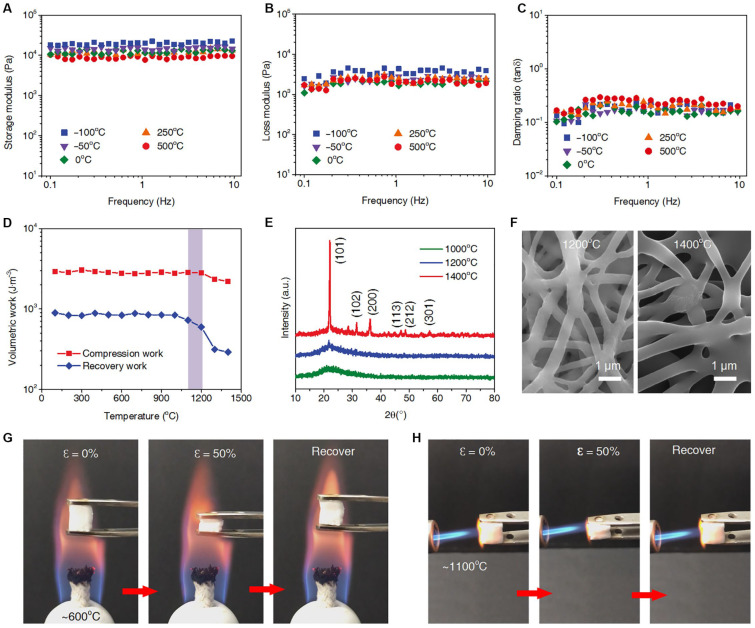
Mechanical properties of the CNFAs over a wide range of temperatures. (**a**–**c**) Storage modulus, loss modulus, and damping ratio of the CNFAs versus angular frequency (0.1 to 10 Hz) at temperatures from −100 °C to 500 °C, with an oscillatory value of 3%. (**d**) Compression and recovery study of the CNFAs after treatment at various temperatures for 30 min. (**e**) XRD patterns of CNFAs after treatment at 1000 °C, 1200 °C, and 1400 °C for 30 min. (**f**) SEM images of CNFAs after treatment at 1200 °C and 1400 °C for 30 min. Compression and recovery process of the CNFAs in the flame of (**g**) an alcohol lamp and (H) a butane blowtorch (adapted with permission from Ref. [103]. OpenAcess).

**Figure 20 gels-09-00208-f020:**
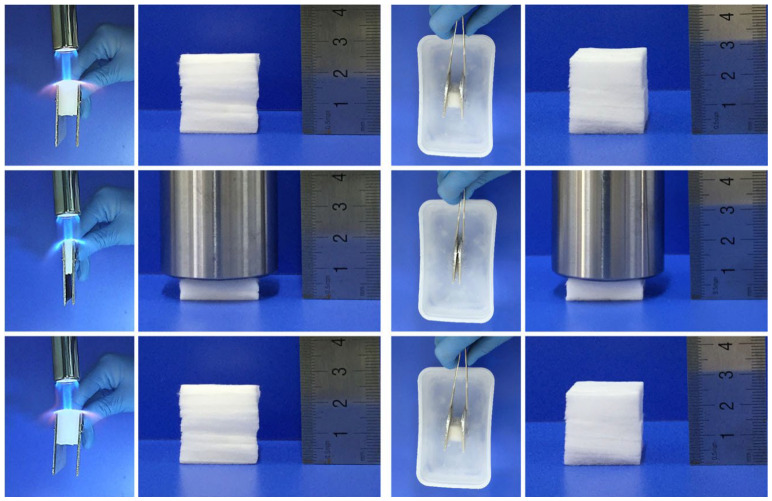
Photographs showing the compressibility and fire resistance of the SAC sponges burned with a butane blowlamp or compressibility immersed in liquid N2 (adapted with permission from Ref. [114]. 2020, The Authors).

**Figure 21 gels-09-00208-f021:**
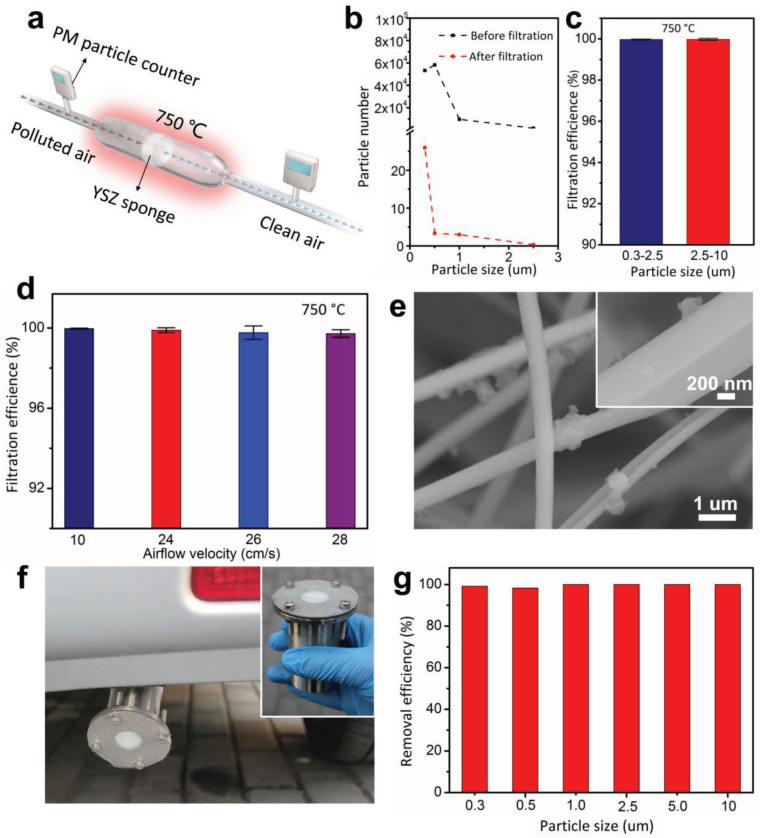
Filtration test at high temperature. (**a**) Schematic of high-temperature PM filtration measurement setup. (**b**) Concentration of PM particles before and after filtration at 750 °C. (**c**) PM0.3–2.5 and PM2.5–10 filtration efficiency at 750 °C. (**d**) Filtration efficiency of PM particles with different airflow velocities at 750 °C. (**e**) SEM images of YSZ nanofiber sponge after filtration at 750 °C. (**f**) PM measurement of automobile exhaust gas with YSZ nanofiber sponge filter. (**g**) Filtration efficiency for different particle sizes of automobile exhaust gas (adapted with permission from Ref. [47]. 2018, WILEY-VCH Verlag GmbH Co. KGaA).

**Figure 22 gels-09-00208-f022:**
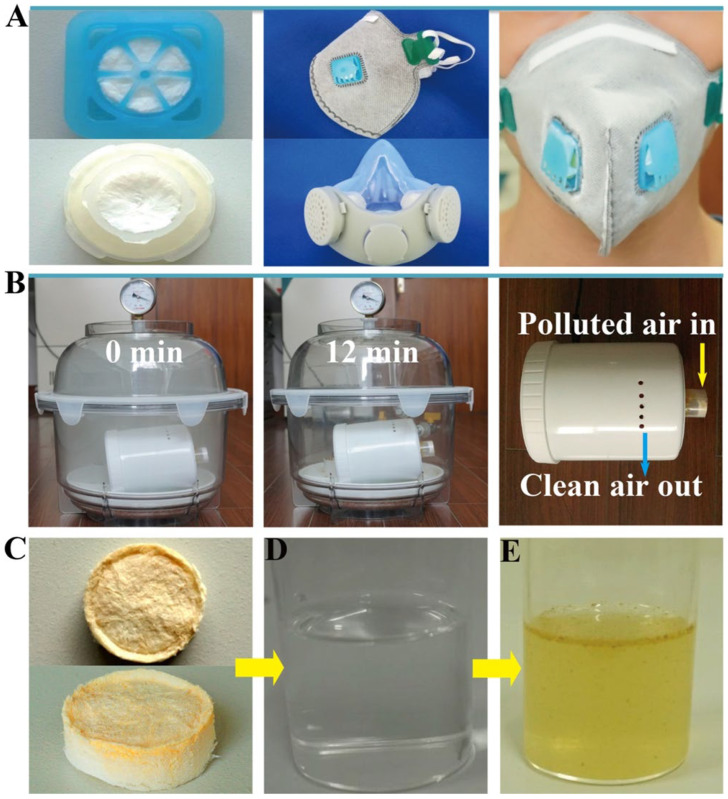
(**A**) Different types of breathing masks using the hydrophobic HAP nanowire aerogel filter. (**B**) Digital images of a hermetic chamber filled with highly polluted air with a very high concentration of PM, and a prototype filter model prepared using the hydrophobic HAP nanowire aerogel for air purification for 12 min. (**C**) Digital images of the hydrophobic HAP nanowire aerogel filter used in the prototype filter model after the air purification process for 2 h. (**D**) Digital image of the hydrochloric acid aqueous solution. (**E**) Digital image of the hydrochloric acid aqueous solution after dissolution of the hydrophobic HAP nanowire aerogel filter used for air purification for 2 h (adapted with permission from Ref. [117]. 2018, American Chemical Society).

**Figure 23 gels-09-00208-f023:**
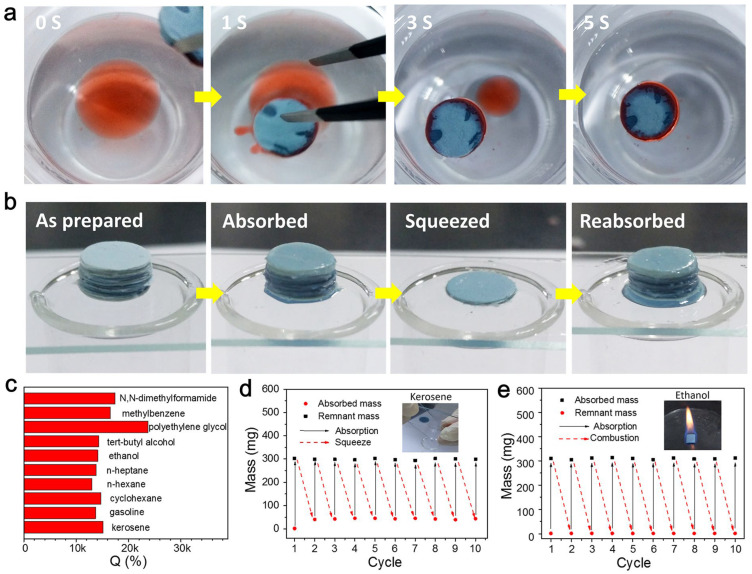
Oil and organic solvent absorption properties of the SiC NWA. (**a**) Absorption process of kerosene (colored with Sudan III for clear presentation) within 5 s. (**b**) Recyclability of the absorption process. (**c**) Absorption capability of the SiC NWA for various organic liquids. (**d**) Recyclability of the SiC NWA for absorption of kerosene when using the squeeze method. (**e**) Recyclability of the SiC NWA for absorption of ethanol when using the direct combustion method (adapted with permission from Ref. [96]. 2018, American Chemical Society).

**Figure 24 gels-09-00208-f024:**
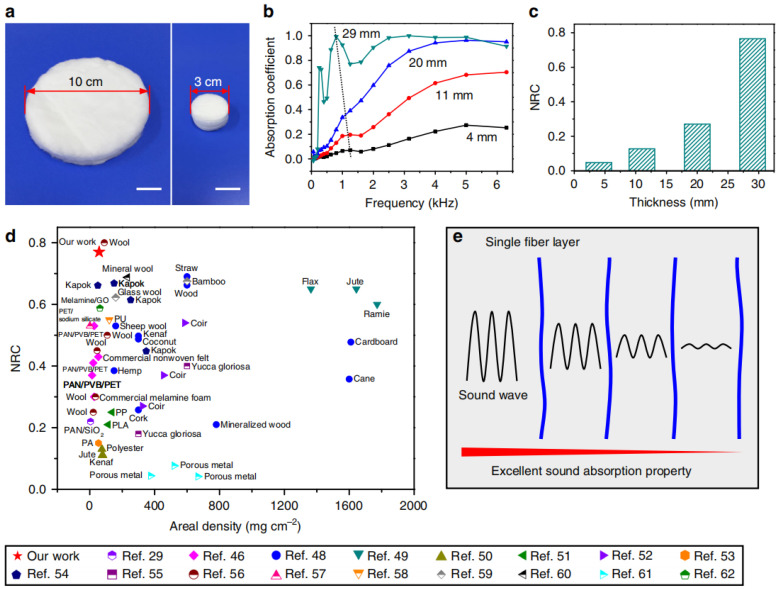
Acoustic absorption properties of the SAC sponges. (**a**) Optical images of the SAC sponges for acoustic absorption property determination. Scale bars, 2 cm. (**b**) Sound absorption coefficient of the SAC sponges with different thicknesses. (**c**) Noise reduction coefficient (NRC) of the SAC sponges with different thicknesses. (**d**) Comparison of the sound absorption properties of our SAC sponges with other sound absorption materials. (**e**) Schematic showing sound transmission though the SAC sponges (adapted with permission from Ref. [114]. 2020, The Authors).

**Figure 25 gels-09-00208-f025:**
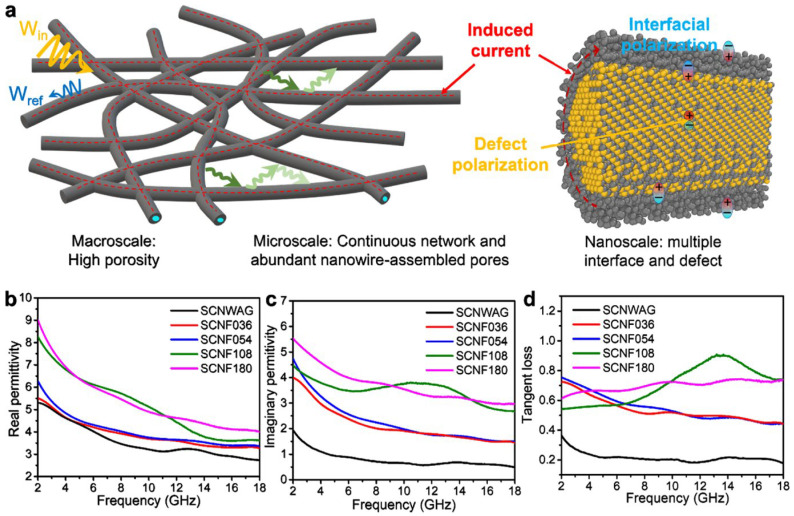
Electromagnetic attenuation mechanism and electromagnetic parameters of SCNWAG and SCNFs. (**a**) Schematic showing the EMW absorption mechanism of the SCNFs. (**b**) Real part of permittivity, (**c**) imaginary part of permittivity, and (**d**) tangent loss of the different samples (adapted with permission from Ref. [123]. 2017, American Chemical Society).

**Figure 26 gels-09-00208-f026:**
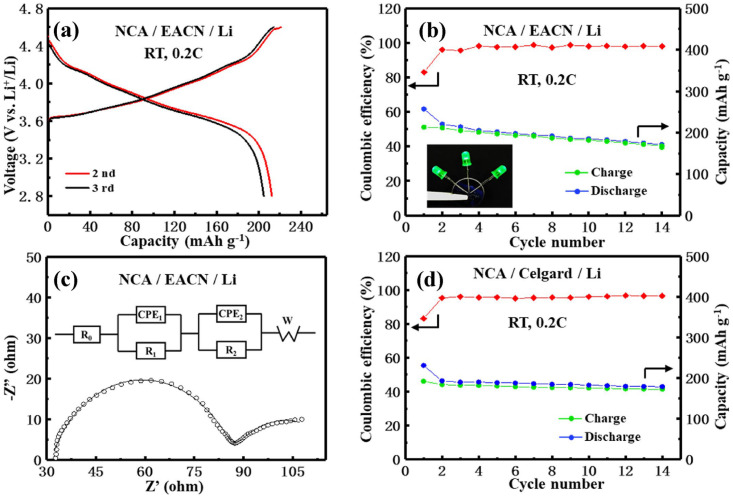
Battery tests and characterizations. All the battery tests were carried out at RT. (**a**) Typical charge/discharge curves of the solid-state NCA/EACN/Li batteries at 0.2C. (**b**) The capacity and Coulombic efficiency of the NCA/EACN/Li batteries at 0.2C. The solid-state battery could efficiently power light-emitting diodes. (**c**) Impedance measurements for the NCA/EACN/Li battery before cycling. The inset figure is the equivalent circuit of the impedance data. (**d**) The capacity and Coulombic efficiency of the liquid NCA/Li batteries at 0.2C (adapted with permission from Ref. [125]. 2019, Elsevier).

**Table 1 gels-09-00208-t001:** Classification and chemical composition of aluminosilicate fibers [1].

	Class Number	1	2	3	4	5	6
Components	Grading temperature/°C	<1200	1260	1400	1400	1550	1600
	SiO_2_	53.9	53	45	55	15	5
	Al_2_O_3_	43.4	47	55	41	85	95
	Cr_2_O_3_	-	-	-	4	-	-
	TiO_2_	1.7	-	-	-	-	-
	Fe_2_O_3_	0.8	-	-	-	-	-
	K_2_O + Na_2_O	0.2	-	-	-	-	-
Phase	Content						
	Mullite	~65	65	75	57	54	18
	Cristobalite	35	35	25	43	-	-
	Al_2_O_3_	-	-	-	-	-	-
	Structure	Amorphous	Amorphous	Amorphous	Amorphous	Polycrystalline	Polycrystalline

**Table 2 gels-09-00208-t002:** Materials and calcination conditions for preparation of ceramic-based nanofibers and aerogels.

Ceramics	Precursors	Solvents	Calcination Conditions	Methods	Products	References
Al_2_O_3_	Aluminum acetate	Ethanol	20 °C·min^−1^, 1000 °C for 2 h, in air	Electrospinning	Nanofiber	[27]
	Aluminum powder	H_2_O	1 °C·min^−1^, 600 °C for 2 h, 5 °C·min^−1^, 700–1000 °C for 2 h, in air	Electrospinning	Nanofiber	[28]
	AlCl_3_·6H_2_O, aluminum powder	H_2_O	4 °C·min^−1^, 600–1100 °C for 2 h, in air	Solution blow spinning	Nanofiber	[31]
	AlCl_3_·6H_2_O, aluminum powder	H_2_O	4 °C·min^−1^, 400–1200 °C for 4 h, in air	Centrifugal spinning	Nanofiber	[23]
	Al(NO_3_)_3_·9H_2_O, aluminum isopropoxide	H_2_O	5 °C·min^−1^, 900 °C for 1 h, in air	Electrospinning	Nanofiber	[100]
	Aluminum isopropoxide	H_2_O	3 °C·min^−1^, 650 °C/1200 °C for 3 h, in air	Centrifugal spinning	Nanofiber	[33]
Mullite	Aluminum trisec-butoxide, polyhydromethylsiloxane	Isopropanol, DMF, ethylacetoacetate	2 °C·min^−1^, 800–1500 °C, in air	Electrospinning	Aerogel	[101]
	Aluminium isopropoxide, Al(NO_3_) _3_·9H_2_O, TEOS	H_2_O, ethanol	800–1400 °C for 2 h, in air	Electrospinning	Nanofiber	[35]
	Aluminum acetate, Colloidal silica	H_2_O, ethanol	5 °C·min^−1^, 800 °C for 1 h, 800–1200 °C for 1 h in air	Electrospinning	Nanofiber	[36]
	Aluminum acetate, TEOS	H_2_O, ethanol	5 °C·min^−1^, 800 °C for 1 h, 1000 °C for 1 h, in air	Electrospinning	Nanofiber	[102]
	TEOS, Al(NO_3_)_3_·9H_2_O	THF	600–1000 °C, in air	Solution blow spinning	Nanofiber	[37]
ZrO_2_	ZrOCl_2_·8H_2_O	H_2_O	1 °C·min^−1^, 800 °C for 1 h, 5 °C·min^−1^, 1200 °C for 1 h, in air	Centrifugal spinning	Nanofiber	[45]
	Zirconium acetate	Acetic acid	200–1000 °C for 2 h, in air	Electrospinning	Nanofiber	[71]
	ZrOCl_2_·8H_2_O	H_2_O	600–1300 °C, in air	Electrospinning	Nanofiber	[72]
	Zirconium acetate hydroxide	DMF	5 °C·min^−1^, 280 °C for 1 h, 1 °C·min^−1^, 800 °C for 3 h, in air	Electrospinning	Nanofiber	[73]
	ZrOCl_2_·8H_2_O	Ethanol, H_2_O	2 °C·min^−1^, 800 °C for 200 min, in air	Solution blow spinning	Nanofiber	[44]
	ZrOCl_2_·8H_2_O	H_2_O_2_, H_2_O	1.2–3 °C·min^−1^, 1300 °C form 3 h, in steam atmosphere	Centrifugal spinning	Nanofiber	[46]
	Zirconium n-propoxide	Ethanol	2 °C·min^−1^, 800 °C for 200 min, in air	Solution blow spinning	Nanofiber	[47]
SiO_2_	TEOS	H_2_O	5 °C·min^−1^, 600–1200 °C, in air	Electrospinning	Nanofiber (82,109–111), Aerogel (108)	[81,103,104,105,106]
	TEOS	H_2_O	5 °C·min^−1^, 800 °C for 2 h, in air	Electrospinning	Aerogel	[83]
	TEOS	Ethanol	6 °C·min^−1^, 550 °C for 1 h, in air	Air-jet spinning	Nanofiber	[107]
	TEOS	Ethanol	850 °C,6 h, in air	Centrifugal jet spinning	Nanofiber	[108,109]
	TEOS	Ethanol	2 °C·min^−1^, 300, 600 and 900 °C, in air	Centrifugal spinning	Nanofiber	[84]
	TEOS	Ethanol	10 °C·min^−1^, 250–1000 °C for 3 h, in air	Electrospinning	Nanofiber	[110]
BN	Boric acid, melamine	H_2_O, tertiary butyl alcohol	1200 °C for 3 h in NH_3_	Self-assembly	Aerogel	[91]
Si_3_N_4_	Methyltrimethoxysilane, dimethyldimethoxysilane	Ethanol	5 °C·min^−1^, 1500 °C for 2 h in N_2_	Chemical vapor deposition	Aerogel	[93]
	Silica sol, carbon black	/	3 °C·min^−1^, 1600 °C for 3 h in N_2_	Chemical vapor deposition	Aerogel	[96]
	PCS	DMF, THF	2 °C·min^−1^, 210 °C for 2 h in air, calcinated at 800 °C for 2 h, 1300 °C for 2 h in Ar	Polymer conversion	Aerogel	[111]
	PCS	Toluene, DMF	Stabilized at 170 °C for 3 h in air, calcinated at 1100–1500 °C in N_2_/calcinated at 1500 °C in Ar	Polymer conversion	Nanofiber	[54]
	Methyltrimethoxysilane, dimethyldimethoxysilane	Ethanol, H_2_O	5 °C·min^−1^, 1550 °C for 2 h, in Ar	Chemical vapor deposition	Aerogel	[87]
	PCS	Xylene, H_2_O	Stabilized at 200 °C for 10 h in air, calcinated at 1400 °C for 2 h in Ar	Polymer conversion	Nanofiber	[55]
	Polysilocarbonsilane	/	Stabilized at 160–220 °C for 6–8 h in air, calcinated at 1800 °C in Ar	Melt spinning	Nanofiber	[112]

## Data Availability

Not applicable.

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
