# Peer review of "Fabrication and Applications of Ceramic-Based Nanofiber Materials Service in High-Temperature Harsh Conditions—A Review"

_gels, 2023, doi:10.3390/gels9030208_

Round 1

Reviewer 1 Report

The authors of the work “Fabrication and applications of ceramic-based nanofiber materials service in high-temperature harsh conditions—A review” have made an attempt to write the existing information on the subject in an organized way.

It is necessary that the authors review the manuscript in detail and rewrite it, since there are many grammatical errors, and this makes it very difficult to read and understand.

Furthermore, it is notable that there is no logical structure of the information along the text. For example, in subsection 2.1 the authirs talk about alumina fibers. This section begins with examples of Alumina, Mullite, and Aluminosilicates. The text begins with the description of Aluminosilicates, then alumina is explained, but then the authors return to the aluminosilicates. It is not possible to understand why the authors used like structure of the text. I suggest that the information in each subsection be organized by the obtaining methods, or by the type of material in ascending order of their properties.

In addition, throughout the text it is constantly noted that there is no connection between the sentences within the same paragraph.

It is necessary that all the abbreviations in the manuscript are deciphered, for example, for the authors, what does PVP mean?

In line 34 it is not clear which three classifications are being talked about, since only two are present in the text.

It is not clear, why in section 2 the authors talk only about the alumina, zirconia, and glass, fibers and there is nothing about the fibers of other types of ceramic materials, for example, carbides, nitrides, and others. This means that such materials do not exist. I consider, that information about such materials must be added.

Alon the manuscript more than 90% of references cited only publications by Chinese researchers, creating an illusion that only China is investigating the issue of manufacturing and applying ceramic nanofibers, but it is no true. It is necessary that the References Section contains at least 50% of references related to research on this topic, carried out by scientific groups from other countries, including the US, Canada, Russia, Spain, Germany, Japan, France and etc.

Moreover, is necessary to include a general table, where the information used in this manuscript is organized and presented in a logical way.

Based on the above, I consider that the manuscript cannot be accepted in its current version.

I recommend the authors to rewrite the manuscript completely so that you get a better presentation and can be read in an understandable and easy way.

I recommend that the authors completely rewrite the manuscript so that this work could be better presented, understandable and read easily.

Author Response

Dear Editor and Reviewers,

Thanks for your consideration for the publication of our work entitled "Fabrication and applications of ceramic-based nanofiber mate-rials service in high-temperature harsh conditions—A review" (Manuscript ID: gels-2225323).

We are very grateful for the detailed and thoughtful comments made by the reviewers. The comments are very helpful for us to further improve our work. The revision has been made to fully address the reviewer’s comments point by point. And some expressions have been correspondingly changed as shown in postil in the manuscript.

The authors of the work “Fabrication and applications of ceramic-based nanofiber materials service in high-temperature harsh conditions—A review” have made an attempt to write the existing information on the subject in an organized way. It is necessary that the authors review the manuscript in detail and rewrite it, since there are many grammatical errors, and this makes it very difficult to read and understand.

We thank the reviewer’s positive assessments and constructive suggestions. A detailed response to each point is provided in the following.

(1)Furthermore, it is notable that there is no logical structure of the information along the text. For example, in subsection 2.1 the authirs talk about alumina fibers. This section begins with examples of Alumina, Mullite, and Aluminosilicates. The text begins with the description of Aluminosilicates, then alumina is explained, but then the authors return to the aluminosilicates. It is not possible to understand why the authors used like structure of the text. I suggest that the information in each subsection be organized by the obtaining methods, or by the type of material in ascending order of their properties.

Response: We thank the reviewer to point out these problems. We have checked our manuscript carefully with section 2.1 reorganized, and the other errors have been fixed in the revised manuscript.

(2)In addition, throughout the text it is constantly noted that there is no connection between the sentences within the same paragraph.

Response: We thank the reviewer to point out these problems. We have checked our manuscript carefully with explanations added to the whole manuscript to explain the paragraph's relationship.

(3)It is necessary that all the abbreviations in the manuscript are deciphered, for example, for the authors, what does PVP mean?

Response: We thank the reviewer for the nice reminder. The whole name of all abbreviations has been added in the place that they first appear in the modified manuscript.

(4)In line 34 it is not clear which three classifications are being talked about since only two are present in the text.

Response: We thank the reviewer for the nice reminder. The introduction has been rewritten to correct this problem.

(5)It is not clear, why in section 2 the authors talk only about the alumina, zirconia, and glass, fibers and there is nothing about the fibers of other types of ceramic materials, for example, carbides, nitrides, and others. This means that such materials do not exist. I consider, that information about such materials must be added.

Response: We thank the reviewer for the nice reminder. We have changed the structure of section 2 with the mentioned fibers as carbides, nitrides, and others added in this manuscript.

(6)Alon the manuscript more than 90% of references cited only publications by Chinese researchers, creating an illusion that only China is investigating the issue of manufacturing and applying ceramic nanofibers, but it is no true. It is necessary that the References Section contains at least 50% of references related to research on this topic, carried out by scientific groups from other countries, including the US, Canada, Russia, Spain, Germany, Japan, France and etc.

Response: We thank the reviewer for the nice reminder. But I could not agree with this point. Although there we listed references, it seemed as if the authors are Chinese authors, and many of them worked in the institution of USA or European. And, Chinese researchers such as Prof. Wu Hui at Tsinghua University, Prof. Ding Bin at Donghua University are the main researchers who focused on this field for decades. Their work should be given attention.

(7)Moreover, is necessary to include a general table, where the information used in this manuscript is organized and presented in a logical way.

Response: We thank the reviewer for the nice reminder. We have added a logic graph for this manuscript as shown in Figure 1.

(8) Based on the above, I consider that the manuscript cannot be accepted in its current version. I recommend the authors to rewrite the manuscript completely so that you get a better presentation and can be read in an understandable and easy way. I recommend that the authors completely rewrite the manuscript so that this work could be better presented, understandable, and read easily.

Response: We thank the reviewer for the nice reminder. We have reorganized this manuscript with English recorrect by the MDPI English editing service.

Reviewer 2 Report

Comments: gels-2225323

The authors presented a brief review of the Fabrication and applications of ceramic-based nanofiber materials service in high-temperature harsh conditions—A review. The review is interesting and well-briefed. The paper has been accepted after minor changes.

·         Abstract should be enhanced with some major results. If possible add the qualitative results.

·         Please add very recent works related to your study.

·         Summary should be elaborated more.

Table 1 has been given reference if it is taken from other sources.

Author Response

Dear Editor and Reviewers,

Thanks for your consideration for the publication of our work entitled "Fabrication and applications of ceramic-based nanofiber mate-rials service in high-temperature harsh conditions—A review" (Manuscript ID: gels-2225323).

We are very grateful for the detailed and thoughtful comments made by the reviewers. The comments are very helpful for us to further improve our work. The revision has been made to fully address the reviewer’s comments point by point. And some expressions have been correspondingly changed as shown in postil in the manuscript.

The authors presented a brief review of the Fabrication and applications of ceramic-based nanofiber materials service in high-temperature harsh conditions—A review. The review is interesting and well-briefed. The paper has been accepted after minor changes.

(1) Abstract should be enhanced with some major results. If possible add the qualitative results.

Response: We thank the reviewer for the nice reminder. The abstract has been rewritten.

(2)Please add very recent works related to your study.

Response: We thank the reviewer for the nice reminder. We have added our related work to the manuscript as shown in references 71-75, 79, 105-106.

(3)Summary should be elaborated more.

Response: We thank the reviewer for the nice reminder. The summary has been rewritten.

(4)Table 1 has been given reference if it is taken from other sources.

Response: We thank the reviewer for the nice reminder. The reference of Table 1 has been added.

Round 2

Reviewer 1 Report

I consider that this manuscript is not yet ready for publication and should be corrected based on the following comments:

1) In section 2.4 appropriate references should be included.

2) A table with the current developments and main findings of various researchers in the field on the synthesis of nanofibers and aerogel ceramics should be added. This table should present the information from the references used in this manuscript and should include the relevant information so that future readers can use it. For example, the table must include the type of material used, whether it is airgel or fiber, the technology used for its manufacture, the parameters used in the production process, main findings, and the importance of the results.

3) In each subsection, a conclusion of the information must be presented, indicating the advantages and disadvantages of the cited references. Also, a forecast of future developments in the studied fields should be included in each subsection.

4) I consider that my previous comment about adding more references of investigations carried out by scientific groups from the US, Canada, Russia, Spain, Germany, Japan, France and etc., and showing new developments and findings in the area of "manufacturing and applying ceramic nanofibers and aerogels” has been ignored.

5) Therefore, I want to check the affiliations in each article presented in this manuscript. For this reason, I ask the authors to add the DOI numbers to each reference in the References section. In addition, in this section you can find only 7 publications related to the author Ding B., and 5 related to the author Wu H., which represents only 6% and 4% of the references, respectively. This is why I ask the authors to cite works carried out in other countries and by scientists no less important than Prof. Wu Hui and Ding Bin.

Once the authors make the necessary corrections, it will be possible to continue with the revision of the manuscript.

Author Response

Dear Editor and Reviewers,

Thanks for your consideration for the publication of our work entitled "Fabrication and applications of ceramic-based nanofiber mate-rials service in high-temperature harsh conditions—A review" (Manuscript ID: gels-2225323).

We are very grateful for the detailed and thoughtful comments made by the reviewers. The comments are very helpful for us to further improve our work. The revision has been made to fully address the reviewer’s comments point by point. And some expressions have been correspondingly changed as shown in postil in the manuscript.

Yours sincerely,

Junxiong Zhang

Reviewer 1

I consider that this manuscript is not yet ready for publication and should be corrected based on the following comments:

We thank the reviewer’s positive assessments and constructive suggestions. A detailed response to each point is provided in the following.

1) In section 2.4 appropriate references should be included.

Response: We thank the reviewer for the nice advice. Some appropriate references have been added in section 2.4.

2) A table with the current developments and main findings of various researchers in the field on the synthesis of nanofibers and aerogel ceramics should be added. This table should present the information from the references used in this manuscript and should include the relevant information so that future readers can use it. For example, the table must include the type of material used, whether it is airgel or fiber, the technology used for its manufacture, the parameters used in the production process, main findings, and the importance of the results.

Response: We thank the reviewer for the nice reminder. We have added a table as shown in Table 2 for introducing the synthesis of ceramic-based nanofibers and aerogel.

3) In each subsection, a conclusion of the information must be presented, indicating the advantages and disadvantages of the cited references. Also, a forecast of future developments in the studied fields should be included in each subsection.

Response: We thank the reviewer for the nice reminder. We’ve added summaries in each subsection in the whole manuscript.

4) I consider that my previous comment about adding more references of investigations carried out by scientific groups from the US, Canada, Russia, Spain, Germany, Japan, France and etc., and showing new developments and findings in the area of "manufacturing and applying ceramic nanofibers and aerogels” has been ignored.

Response: We thank the reviewer for the nice reminder. The research of foreign groups in the previous version has been shown in references 3, 7-13, 20, 22, 28, 33-35, 37, 49, 66, 72, 74, 90, and 92. And, we do add more research as shown in the new version references 57-58, 79, 81, and 85.

5) Therefore, I want to check the affiliations in each article presented in this manuscript. For this reason, I ask the authors to add the DOI numbers to each reference in the References section. In addition, in this section you can find only 7 publications related to the author Ding B., and 5 related to the author Wu H., which represents only 6% and 4% of the references, respectively. This is why I ask the authors to cite works carried out in other countries and by scientists no less important than Prof. Wu Hui and Ding Bin.

Response: We thank the reviewer for the nice reminder. We have added the DOI of each reference in the references. The mainly Chinese research groups of nanofibers and aerogel include the following groups of Pro.f Wu H (4), Pro.f Ding B (5), Pro.f Wang H (12), Pro.f Zhu M (3), Pro.f Cui S (4), Pro.f Yang J (8), Pro.f Chen Z (6), whose work displayed a ratio of 37.1%. The foreign groups showed a ratio of 18.6% not including the Chinese-Foreign researcher as Pro.f Duan X and others. On the other hand, we added some research as shown in the new version references 57-58, 79, 81, and 85.

Round 3

Reviewer 1 Report

From table 2 “Materials and calcination conditions for preparation of ceramic-based nanofibers and aerogel” it is not clear why reference [6] is indicated. Is it perhaps a table that summarizes the works of reference [6]?

Table 2 should contain a summary of the information presented in this manuscript. For this reason, it is necessary to present the information of the links used here, which are about 140.

Furthermore, from Table 2 it is not clear, which works are dedicated to the production of nanofibers or aerogels. It is necessary to add an additional column, in where it is indicated which works refer to nanofibers and which to aerogels.

 It is necessary to correct the English of the manuscript again. For example, I believe that the authors instead of the term “oxidation fibers” meant “Oxide Fibers”.

Subsections “3.2 Nitride aerogels” and “3.3 Carbide aerogels” do not have the required summaries. Please add them.
